# DiTTo-TTS: Diffusion Transformers for Scalable Text-to-Speech without Domain-Specific Factors

**Keon Lee**[1], **Dong Won Kim**[1], **Jaehyeon Kim**[2][\*] **Seungjun Chung**[1], **Jaewoong Cho**[1]
[1]KRAFTON, [2]NVIDIA

## Abstract

Large-scale latent diffusion models (LDMs) excel in content generation across various modalities, but their reliance on phonemes and durations in text-to-speech (TTS) limits scalability and access from other fields. While recent studies show potential in removing these domain-specific factors, performance remains suboptimal. In this work, we introduce DiTTo-TTS, a Diffusion Transformer (DiT)-based TTS model, to investigate whether LDM-based TTS can achieve state-of-the-art performance without domain-specific factors. Through rigorous analysis and empirical exploration, we find that (1) DiT with minimal modifications outperforms U-Net, (2) variable-length modeling with a speech length predictor significantly improves results over fixed-length approaches, and (3) conditions like semantic alignment in speech latent representations are key to further enhancement. By scaling our training data to 82K hours and the model size to 790M parameters, we achieve superior or comparable zero-shot performance to state-of-the-art TTS models in naturalness, intelligibility, and speaker similarity, all without relying on domain-specific factors. Speech samples are available at `https://ditto-tts.github.io`.

## 1 Introduction

Diffusion models have demonstrated impressive generative abilities in a wide range of tasks, including image (Ho et al., 2020; Song et al., 2020b), audio (Chen et al., 2020; Kong et al., 2020), and video (Singer et al., 2022; Bar-Tal et al., 2024) generation. Building on these advancements, latent diffusion models (LDMs) (Rombach et al., 2022) enable more efficient learning (Chen et al., 2023a; Peebles & Xie, 2023; Blattmann et al., 2023; Liu et al., 2023). These models leverage latent representations to capture the essential features of input data in lower dimensions, significantly reducing computational costs while preserving the quality of the generated outputs.

There has been a proliferation of LDM-based text-to-speech (TTS) models (Shen et al., 2024; Ju et al., 2024). One challenge in TTS is achieving temporal alignment between the generated speech and the input text, meaning that each word or phoneme must synchronize with the corresponding speech audio at the correct time (Kim et al., 2020; Popov et al., 2021). To address this challenge, prior LDM-based models employ domain-specific factors, such as phonemes and durations, which complicates data preparation and hinders the scaling of datasets for training large-scale models (Gao et al., 2023). Inspired by the original approach of LDMs in other domains—where alignment occurs without domain-specific factors—this limitation prompts the use of pre-trained text and speech encoders that align text and speech solely through cross-attention mechanisms, eliminating the need for phoneme and duration predictions and simplifying the model architecture. Recent pioneering works (Gao et al., 2023; Lovelace et al., 2024) demonstrate the potential of this approach, though their performance remains suboptimal. This raises the following questions:

> Can LDM truly achieve state-of-the-art TTS performance at scale without relying on domain-specific factors, as in other fields? If so, what key aspects drive this success?

---

[\*]This work was done when Jaehyeon Kim was at KRAFTON.
Email: <keonlee@krafton.com>. Correspondence: <jwcho@krafton.com>.

To address this question, we conduct a rigorous investigation into the aspects contributing to sub-optimal performance and resolve these issues through comprehensive experiments, demonstrating that achieving competitive performance is indeed possible. We highlight three key aspects of our approach: First, we demonstrate that the Diffusion Transformer (DiT) (Peebles & Xie, 2023) is more suitable for TTS tasks than U-Net (Ronneberger et al., 2015) architectures, identifying minimal modifications such as long skip connections and global adaptive layer normalization (AdaLN) (Chen et al., 2024). By successfully transplanting DiT to TTS, we introduce DiTTo-TTS (or simply DiTTo). Second, we show that modeling variable speech length improves performance compared to the fixed length modeling used in previous works (Gao et al., 2023; Lovelace et al., 2024). Instead of using padding with fixed length modeling, we introduce Speech Length Predictor module that predicts the total speech length during inference based on the given text and speech prompt. Lastly, we explore the crucial conditions for effective latent representation in LDM-based TTS models, including semantic alignment, which can be achieved by either using a text encoder jointly trained with speech data or a speech autoencoder incorporating an auxiliary language modeling objective.

Our extensive experiments show that our model, which does not rely on speech domain-specific factors, achieves superior or comparable performance compared to existing state-of-the-art TTS models (Casanova et al., 2022; Wang et al., 2023; Le et al., 2023; Kim et al., 2024a) (see Section 4 for all baselines). This is demonstrated in both English-only and multilingual evaluations across naturalness, intelligibility, and speaker similarity. Notably, the base-sized DiTTo outperforms a state-of-the-art autoregressive model (Kim et al., 2024a), offering 4.6 times faster inference while using 3.84 times less model size. Additionally, we show that our model scales effectively with increases in both data and model sizes.

## 2 RELATED WORK

**Latent Diffusion Models (LDMs)**    LDM (Rombach et al., 2022) improves modeling efficiency of the diffusion model (Ho et al., 2020; Song et al., 2021) by operating in a latent space, achieving remarkable performance in generating realistic samples. Initially applied in image generation, their success is attributed to the reduced dimensionality of the latent space, facilitating efficient training and sampling (Rombach et al., 2022). Notably, guided diffusion (Dhariwal & Nichol, 2021; Ho & Salimans, 2021) has been expanded to various applications of LDMs, such as image editing (Brooks et al., 2023) and image retrieval (Gu et al., 2023). In the field of audio signals, techniques such as style transfer, inpainting, and super-resolution have been explored, along with text-guided audio and speech generation (Liu et al., 2023; Lee et al., 2024). In the context of TTS, however, applying LDMs to TTS (Shen et al., 2024; Ju et al., 2024) necessitates domain-specific elements such as phonemes, phoneme-level durations. This is primarily due to the need for precise text-speech alignment and higher audio fidelity requirements.

**Text-to-Speech without Domain-Specific Factors**    Recent diffusion-based works, such as Simple-TTS (Lovelace et al., 2024) and E3 TTS (Gao et al., 2023), pioneer the exploration of non-autoregressive, U-Net-based TTS models without relying on complex processes like phonemization and phone-level duration prediction. While these models demonstrate the potential of simplified frameworks, they impose a fixed length on the target audio, which not only restricts the flexibility of the generated audio length but also reduces quality due to the need for padding predictions. In Simple-TTS (Lovelace et al., 2024), random-length padding, representing the difference between the maximum fixed length and the target audio length, is included in the loss calculation. In E3 TTS (Gao et al., 2023), the loss for padding frames is weighted at $\frac{1}{10}$ of that for non-padding frames. The most relevant concurrent works to our paper are SimpleSpeech (Yang et al., 2024b) and E2 TTS (Eskimez et al., 2024). SimpleSpeech employs an LLM for speech length prediction and scalar quantization-based speech codec for diffusion modeling, while E2 TTS, building on Voicebox (Le et al., 2023), simplifies text conditioning and removes the need for explicit alignment.

We provide a discussion on another concurrent work, Seed-TTS (Anastassiou et al., 2024), and related work on large-scale TTS and neural audio codec in Appendix A.1.

## 3 METHOD

In this section, we present our method in the following order: Section 3.1 introduces the preliminary concepts necessary for understanding our approach and explains their connection to our work. Section 3.2 outlines the specifics of the proposed method and the training process. Finally, Section 3.3 provides a detailed description of the model architecture and its components.

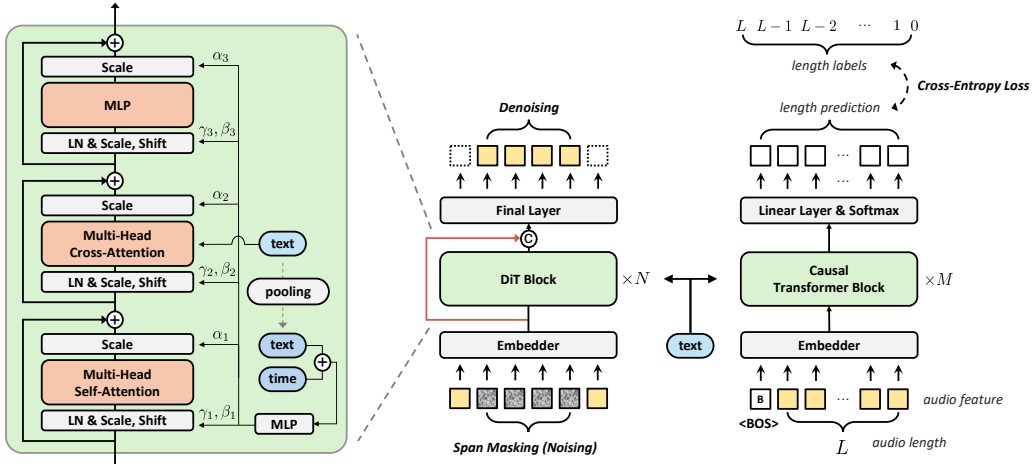

Figure 1: **An overview of DiTTo-TTS**. (*middle*) The LDM backbone is trained to denoise a span-masked noisy segment given its contextual counterpart, without utilizing phoneme and phoneme-level duration (see Section 3.1). (*left*) The inner structure of the DiT block incorporates multi-head cross-attention with global AdaLN (see Section 3.3). (*right*) The speech length predictor is based on causal transformer blocks (see Section 3.2). Both DiT blocks and the speech length predictor employ cross-attention to condition on text representation. Additionally, DiT blocks utilize AdaLN with the mean-pooled text embedding. '**+**' denotes addition, and '**c**' represents concatenation. '**text**' in sky blue oval represents the embedding from text encoder. The red line indicates a long skip connection between the states before and after the DiT blocks.

## 3.1 PRELIMINARY

Diffusion models (Ho et al., 2020; Song et al., 2021) are a class of generative models that iteratively transform a simple noise distribution into a complex data distribution through a stochastic denoising process. They define a forward process that progressively adds Gaussian noise to the input data as time step increases. The reverse generative process then estimates the added noise to reconstruct the original data. Conditional diffusion models enhance this framework by incorporating additional information, such as text descriptions in text-to-image generation (Dhariwal & Nichol, 2021; Saharia et al., 2022) or phonemes and their durations in TTS (Popov et al., 2021; Kim et al., 2022). While diffusion models can operate directly on real-world data, many of them are applied in the latent space (Rombach et al., 2022; Blattmann et al., 2023; Peebles & Xie, 2023; Ghosal et al., 2023). Thanks to the reduced dimensionality, this approach improves computational efficiency and output quality by allowing diffusion models to focus on the semantic information of the data while the autoencoder handles the high-frequency details that are less perceptible (Rombach et al., 2022).

In our setting, a conditional LDM can be formulated as follows. Given speech audio, an autoencoder produces its latent representation $z_{speech}$, and the diffusion model is trained to predict $z_{speech}$ at each diffusion step $t \in [1, T]$ conditioned on a text token sequence $x$. Specifically, the noised latent $z^{(t)}$ is expressed as $\alpha_t z_{speech} + \sigma_t \epsilon$, where $\epsilon$ is sampled from the standard normal distribution and $\alpha_t$ and $\sigma_t$ are defined by a noise schedule. Note that $z^{(1)}$ is $z_{speech}$ and $z^{(T)}$ follows the standard normal distribution. We use $v$-prediction (Salimans & Ho, 2022) as our model output $v_\theta(z^{(t)}, x, t)$, which predicts $v^{(t)} := \alpha_t \epsilon - \sigma_t z^{(t)}$. This setup provides a mean squared error objective as the training loss:

$$\mathcal{L}_{\text{diffusion}} = \mathbb{E}_{t \sim \mathcal{U}(1,T), \epsilon \sim \mathcal{N}(0,I)} \left[ \|v^{(t)} - v_\theta(z^{(t)}, x, t)\|^2 \right].$$

To enrich the contextual information and facilitate zero-shot audio prompting, we incorporate a random span masking into the model training following (Le et al., 2023; Vyas et al., 2023). We input $z_{mask}^{(t)} := m \odot z^{(t)} + (1 - m) \odot z_{speech}$ to the model, where $\odot$ indicates element-wise multiplication and the binary masking $m$ fully masks with the probability of 0.1 or partially masks a random contiguous segment whose size is between 70% and 100% of data. We also use the binary

span masking as an additional input to the model. This allows the model to explicitly identify which part needs to be generated. The inclusion of masking modifies the training loss to:

$$\mathcal{L}_{\text{diffusion}} = \mathbb{E}_{t \sim \mathcal{U}(1,T), \boldsymbol{\epsilon} \sim \mathcal{N}(0,I)} \left[ \| \boldsymbol{m} \odot (\boldsymbol{v}^{(t)} - \boldsymbol{v}_\theta(\boldsymbol{z}_{mask}^{(t)}, \boldsymbol{m}, \boldsymbol{x}, t)) \|^2 \right]. \tag{1}$$

## 3.2 MODEL AND TRAINING

An overview of our proposed method is presented in Figure 1.

**Text Encoder** We employ a text encoder from a pre-trained encoder-decoder-based large language model (Xue et al., 2022) $p_\phi$, which is parameterized by $\phi$. The model was pre-trained to maximize the log-likelihood of the text token sequence $\log p_\phi(\boldsymbol{x})$. The parameters of the model are kept frozen while training the diffusion model for TTS. We denote the output of the text encoder by $\boldsymbol{z}_{text}$.

**Neural Audio Codec** A neural audio codec, which is parameterized by $\psi$, comprises of three components: 1) an encoder that maps a speech into a sequence of latent representations $\boldsymbol{z}_{speech}$; 2) a vector quantizer converting the latent vector into the discrete code representation; and 3) a decoder that reconstructs the speech from a sequence of the quantized latent representations $\hat{\boldsymbol{z}}_{speech}$. Since alignment between text and speech embeddings is crucial for TTS, we fine-tuned the neural audio codec by aligning with the encoder output of a pre-trained language model. We introduce a learnable linear projection $f(\cdot)$ to match the dimension of $\boldsymbol{z}_{speech}$ to the language model's hidden space. This projected embedding is then used in place of the pre-trained text encoder's embedding within the cross-attention mechanism of the language model's decoder. The neural audio codec is fine-tuned with auxiliary loss that infuse semantic content into the generated representations:

$$\mathcal{L}(\psi) = \mathcal{L}_{NAC}(\psi) + \lambda \mathcal{L}_{LM}(\psi), \quad \mathcal{L}_{LM}(\psi) = -\log p_\phi(\boldsymbol{x} | f(\boldsymbol{z}_{speech})), \tag{2}$$

where $\mathcal{L}_{NAC}(\psi)$ represents the loss function used during the pre-training of the neural audio codec (Kim et al., 2024a). The parameter $\lambda$ controls the influence of $\mathcal{L}_{LM}$, with $\lambda = 0$ indicating the pre-training phase. When $\lambda > 0$, a pre-trained language decoder performs causal language modeling on the text token sequences $\boldsymbol{x}$, conditioned on the speech latent vector $\boldsymbol{z}_{speech}$. While the parameters of the language model decoder remain fixed, gradients are backpropagated to update the linear mappings $f(\boldsymbol{z}_{speech})$. This training strategy aligns the speech latents with the linguistic latents of the pre-trained language model during the autoencoding process.

**Diffusion Model** We are given text embedding $\boldsymbol{z}_{text}$ and speech embedding $\boldsymbol{z}_{speech}$. We train the diffusion model $\mathbf{v}_\theta(\cdot)$ using the objective in Eq. (1), replacing $\boldsymbol{x}$ with $\boldsymbol{z}_{text}$:

$$\mathcal{L}_{\text{diffusion}} = \mathbb{E}_{t \sim \mathcal{U}(1,T), \boldsymbol{\epsilon} \sim \mathcal{N}(0,I)} \left[ \| \boldsymbol{m} \odot (\boldsymbol{v}^{(t)} - \boldsymbol{v}_\theta(\boldsymbol{z}_{mask}^{(t)}, \boldsymbol{m}, \boldsymbol{z}_{text}, t)) \|^2 \right], \tag{3}$$

where $\boldsymbol{z}_{mask}^{(t)} = \boldsymbol{m} \odot \boldsymbol{z}^{(t)} + (1 - \boldsymbol{m}) \odot \boldsymbol{z}_{speech}$ is the masked input latent, $\boldsymbol{m}$ is the binary span masking, and $t$ is the diffusion time step. We apply classifier-free guidance (CFG) (Ho & Salimans, 2021) and adopt the diffusion noise schedule from (Lovelace et al., 2024).

**Speech Length Predictor** We introduce a model designed to predict *the total length* of a generated speech for a given text rather than to estimate each phoneme's duration, and the input noise of diffusion model is set by the length from the speech length predictor at inference time. As shown in Figure 1, we employ an encoder-decoder transformer for the speech length predictor. The encoder processes text input bidirectionally to capture comprehensive textual context, while the decoder, equipped with causal masking to prevent future lookahead, receives an audio token sequence from the encoder of the neural codec for speech prompting at inference time. We use cross-attention mechanisms to integrate text features from the encoder, and apply softmax activation in the final layer to predict the number of tokens to be generated within the given maximum length $N$ for sampling purposes. Specifically, the ground truth label for the remaining audio length decreases by one at each subsequent time step, allowing for the acceptance of a speech prompt during inference. The model is trained separate from the diffusion model, using the cross-entropy loss function.

## 3.3 MODEL ARCHITECTURE

We conduct a comprehensive model architecture search to identify the most suitable latent diffusion-based model for TTS, resulting in the adoption of the Diffusion Transformer (DiT) (Peebles & Xie, 2023) model (see Section 5). In doing so, we also integrate recent architectural advancements for

transformer variants, such as the gated linear unit with GELU activation (Shazeer, 2020), rotary position embeddings (Su et al., 2024), and cross-attention with global adaptive layer normalization (AdaLN) (Chen et al., 2024; 2023a). By modeling only the overall speech length, DiTTo learns the detailed duration of each token within the cross-attention layers, enabling the generation of natural and coherent speech. For the latent space, we employ Mel-VAE (Kim et al., 2024a) which is able to compress audio sequences approximately 7-8 times more than EnCodec (Défossez et al., 2023), resulting in a 10.76 Hz code with high audio quality. Details on Mel-VAE and model configurations are in Appendix A.5. We also provide details on our noise scheduler and CFG in Appendix A.6, and through hyperparameter search, we determine the optimal noise and CFG scales to be 0.3 and 5.0, respectively. These settings are fixed throughout the paper unless otherwise specified.

## 4 EXPERIMENTAL SETUP

**Dataset** We employ publicly available speech-transcript datasets totaling 82K hours from over 12K unique speakers across nine languages: English, Korean, German, Dutch, French, Spanish, Italian, Portuguese, and Polish. We train two models: (1) DiTTo-en, a model trained on 55K hour English-only dataset, and (2) DiTTo-multi, a multilingual model trained on 82K hour datasets. Details of each dataset are provided in Appendix A.2. We follow the data preprocessing methodology described in (Kim et al., 2024a), except that we include all samples without any filtering and exclude speaker metadata from the text prompts. It enables the on-the-fly processing of data with different sampling rates at a uniform rate of 22,050 Hz by approximating audio resampling through adjusting the hop size and FFT size according to the ratio of the original sampling rate to 22,050 Hz. We use the same datasets for the speech length predictor with additional LibriSpeech (Panayotov et al., 2015) sets: *train-clean-100*, *train-clean-360*, and *train-other-500*. We find that it helps minimize reliance on text normalization (see Appendix A.2 for discussion). For the evaluation of DiTTo-en and baseline models, we use the *test-clean* subset of LibriSpeech, which consists of speech clips ranging from 4 to 10 seconds with transcripts. For DiTTo-multi, we randomly select 100 examples from the test set of each language dataset, with clip durations ranging from 4 to 20 seconds.

**Training** Following DiT (Peebles & Xie, 2023), we train four different sizes of DiTTo: Small (S), Base (B), Large (L), and XLarge (XL). All models are trained on 4 NVIDIA A100 40GB GPUs, and use $T = 1,000$ discrete diffusion steps. The S and B models of DiTTo-en are trained with a maximum token size of 5,120 and a gradient accumulation step of 2 over 1M steps. The L and XL models are trained with a maximum token size of 1,280 and a gradient accumulation step of 4 over 1M steps. The DiTTo-multi model is trained only in the XL configuration, with a maximum token size of 320 and a gradient accumulation step of 4 over 1M steps. The trainable model parameters for DiTTo-en S, B, L, and XL are 41.89M, 151.58M, 507.99M, and 739.97M, respectively, and 790.44M for DiTTo-multi. For the text encoder, we employ SpeechT5 (Ao et al., 2022) [1] (as in VoiceLDM (Lee et al., 2024)) and ByT5 (Xue et al., 2022) in DiTTo-en and DiTTo-multi, respectively. We use the AdamW optimizer (Loshchilov & Hutter, 2019) with the learning late of 1e-4, beta values of (0.9, 0.999), and a weight decay of 0.0. We use a cosine learning rate scheduler with a warmup of 1K steps. Further details of the speech length predictor is provided in Appendix A.7.

**Inference** To generate speech, we first input the text and speech prompt into the speech length predictor, which determines the total length of the speech $L$ by adding speech prompt length with the predicted length. Further details about the speech length predictor can be found in Appendix A.7. The diffusion backbone then generates the latent speech of length $L$ using the same text and speech prompt pair. The generated latent is decoded into mel-spectrograms using the Mel-VAE decoder, and then converted to raw waveform using a pre-trained vocoder, BigVGAN (Lee et al., 2023b) [2]

**Metrics** We use following objective metrics to evaluate the models: Character Error Rate (CER), Word Error Rate (WER), and Speaker Similarity (SIM) following the procedure outlined in VALL-E (Wang et al., 2023) and CLaM-TTS (Kim et al., 2024a). CER and WER indicate how intelligible and robust the model is, while SIM represents how well the model captures the speaker's identity. For subjective metrics, we employ Similarity MOS (SMOS) to measure the speaker similarity between the prompt and the generated speech, and Comparative MOS (CMOS) to assess relative naturalness and audio quality. Details of each metric and evaluation can be found in Appendix A.3.

---

[1] We use TTS fine-tuned SpeechT5 model from `https://huggingface.co/microsoft/speecht5_tts`.

[2] We use *bigvgan_22khz_80band*.

**Baselines**   We compare the proposed model with state-of-the-art TTS models including (1) *autoregressive models*: VALL-E (Wang et al., 2023), SPEAR-TTS (Kharitonov et al., 2023), CLaM-TTS (Kim et al., 2024a); (2) *non-autoregressive models*: YourTTS (Casanova et al., 2022), Voicebox (Le et al., 2023); and (3) *simple diffusion-based models*: Simple-TTS (Lovelace et al., 2024). Since Voicebox (Le et al., 2023), VALL-E (Wang et al., 2023), NaturalSpeech 2 (Shen et al., 2024), NaturalSpeech 3 (Ju et al., 2024), Mega-TTS (Jiang et al., 2023), and E3 TTS (Gao et al., 2023) are not publicly available, we bring the scores reported in the respective paper or samples provided in demo page. Please refer to Appendix A.4 for more baselines and their details.

**Tasks**   We evaluate our model on two tasks: 1) *continuation*: given a text and an initial 3-second segment of corresponding ground truth speech, the task is to synthesize seamless subsequent speech that reads the text in the style of the provided speech segment; 2) *cross-sentence*: given a text, a 3-second speech segment, and its corresponding transcript (which differs from the given text), the task is to synthesize speech that reads the text in the style of the provided 3-second speech.

## 5   EXPERIMENTAL RESULTS ADDRESSING OUR RESEARCH QUESTIONS

In this section, we present experimental results to answer our two primary research questions:

- **RQ1**: Can LDM achieve state-of-the-art performance in text-to-speech tasks at scale without relying on domain-specific factors, similar to successes in other domains?

- **RQ2**: If LDM can achieve this, what are the primary aspects that enable its success?

We structure our analysis to first demonstrate that LDM can achieve state-of-the-art performance without the need for domain-specific factors (Section 5.1), and then identify the key aspects that contribute to this success (Section 5.2).

### 5.1   WE CAN ACHIEVE STATE-OF-THE-ART PERFORMANCE WITHOUT DOMAIN-SPECIFIC FACTORS (**RQ1**)

**Comparisons with State-of-the-Art Baselines**   For *English-only Evaluation*, we evaluate the performances of DiTTo-en across *continuation* and *cross-sentence* tasks. Recall that DiTTo-en is the model trained on 55K English-only dataset. Following the evaluation setting in (Wang et al., 2023), we employ a subset of the LibriSpeech test-clean dataset. This subset comprises speech clips ranging from 4 to 10 seconds, each with a corresponding transcript. Details on baseline are in Appendix A.4.

Table 1: Performances for the English-only *continuation* task. The boldface indicates the best result, the underline denotes the second best, and the asterisk denotes the score reported in the baseline paper. Inference Time indicates the generation time of 10s speech, and #Param. refers to the number of learnable parameters (model size). The number of diffusion steps (NFE) is 25. The dagger symbol (†) denotes a more extensively trained model with the DDIM sampling schedule (Song et al., 2020a).

| Model | Objective Metrics | | | | | |
| | WER ↓ | CER ↓ | SIM-o ↑ | SIM-r ↑ | Inference Time ↓ | #Param. |
|---|---|---|---|---|---|---|
| Ground Truth | 2.15 | 0.61 | 0.7395 | - | n/a | n/a |
| YourTTS | 7.57 | 3.06 | 0.3928 | - | - | - |
| VALL-E | 3.8* | - | 0.452* | 0.508* | ∼6.2s* | 302M |
| Voicebox | 2.0* | - | 0.593* | 0.616* | ∼6.4s* (64 NFE) | 364M |
| CLaM-TTS | 2.36* | 0.79* | 0.4767* | 0.5128* | 4.15s* | 584M |
| Simple-TTS | 3.86 | 2.24 | 0.4413 | 0.4668 | 17.897s (250 NFE) | 243M |
| DiTTo-en-S | 2.01 | 0.60 | 0.4544 | 0.4935 | **0.884s** | 42M |
| DiTTo-en-B | 1.87 | 0.52 | 0.5535 | 0.5855 | 0.903s | 152M |
| DiTTo-en-L | 1.85 | 0.50 | 0.5596 | 0.5913 | 1.479s | 508M |
| DiTTo-en-XL | **1.78** | **0.48** | 0.5773 | 0.6075 | 1.616s | 740M |
| DiTTo-en-XL† | 1.80 | 0.48 | **0.6051** | **0.6283** | - | 740M |

Table 1 and 2 present the results of the *continuation* and *cross-sentence* tasks, respectively. Our model demonstrates excellent performance across all measures, consistently ranking either first or second. Specifically, the DiTTo-en base (B) model outperforms CLaM-TTS, a state-of-the-art autoregressive model, in terms of naturalness, intelligibility, and speaker similarity, while achieving an inference speed that is 4.6 times faster with 3.84 times smaller model size (even when the parameter

Table 2: Performances for the English-only *cross-sentence* task.

| Model | WER ↓ | CER ↓ | SIM-o ↑ | SIM-r ↑ |
|---|---|---|---|---|
| YourTTS | 7.92 (7.7*) | 3.18 | 0.3755 (0.337*) | - |
| VALL-E | 5.9* | - | - | 0.580* |
| SPEAR-TTS | - | 1.92* | - | 0.560* |
| Voicebox | **1.9*** | - | **0.662*** | **0.681*** |
| CLaM-TTS | 5.11* | 2.87* | 0.4951* | 0.5382* |
| Simple-TTS | 4.09 (3.4*) | 2.11 | 0.5026 | 0.5305 (0.514*) |
| DiTTo-en-S | 3.07 | 1.08 | 0.4984 | 0.5373 |
| DiTTo-en-B | 2.74 | 0.98 | 0.5977 | 0.6281 |
| DiTTo-en-L | 2.69 | 0.91 | 0.6050 | 0.6355 |
| DiTTo-en-XL | 2.56 | **0.89** | 0.6270 | 0.6554 |
| DiTTo-en-XL† | 2.64 | 0.94 | 0.6538 | 0.6752 |

size is doubled and inference steps are increased to 64, DiTTo-en-XL remains faster at 3.715 seconds compared to Voicebox's 6.4 seconds, primarily due to the use of target sequences with shorter lengths.). We also demonstrate further performance improvements by using a more trained model (4.3M training steps) along with the DDIM (Song et al., 2020a) sampling schedule (indicated by †). Our model achieves performance comparable to more complex a non-autoregressive state-of-the-art TTS model which use a phoneme-level duration modeling. Additionally, DiTTo-en surpasses a simple diffusion-based model, Simple-TTS, further demonstrating its effectiveness. We summarize the results of comparison with open-source models in Appendix A.8.

Table 3: Human evaluations on *cross-sentence* task with 40 LibriSpeech test-clean samples (per speaker) show DiTTo-en-XL surpasses the baseline in quality, intelligibility, similarity, and naturalness, nearing Ground Truth. SMOS scores include a 95% confidence interval. The boldface indicates the best result and *recon* indicates the reconstructed audio by Mel-VAE followed by vocoder.

| Model | SMOS | CMOS |
|---|---|---|
| Simple-TTS | 2.15±0.19 | -1.64 |
| CLaM-en | 3.42±0.16 | -0.52 |
| DiTTo-en-XL | **3.91±0.16** | **0.00** |
| Ground Truth (*recon*) | 4.07±0.14 | +0.11 |
| Ground Truth | 4.08±0.14 | +0.13 |

Table 3 presents the results of subjective evaluations. DiTTo-en significantly outperforms the baseline models, Simple-TTS and CLaM-en, and achieves performance comparable to the ground truth in terms of speaker similarity (as measured by SMOS) as well as naturalness, quality, and intelligibility (as assessed by CMOS). In Appendix A.9, Table 12 shows the results of the subjective evaluation compared to the baseline demo samples. We use the provided voice samples as is (as detailed down in Appendix A.4), and our model operates at 22,050 Hz. Our model surpasses all models except NaturalSpeech 3 in SMOS and all models except Voicebox in CMOS. Additionally, we use high-intensity samples from the RAVDESS (Livingstone & Russo, 2018) dataset to test in extreme zero-shot scenarios with out-of-domain prompts. Details about the additional baselines, experimental settings, and results are in Appendix A.4 and Appendix A.11. Comparison results with concurrent works, using our model downsampled to 16,000 Hz, are in Appendix A.10.

For *Multilingual Evaluations*, we provide the result in Appendix A.12 due to space constraints. Table 16 presents the result of the *continuation* task for DiTTo-multi. DiTTo-multi shows better or comparable performances compared to CLaM-TTS performances, as reported in Table 17.

**Scaling Model and Data Size** We train 4 models of different size on 55K hour English-only datasets, which are referred to as small (S), base (B), Large (L), and XLarge (XL). Detailed model configurations are provided in Appendix A.5. Performance improves with model size, as demonstrated in Table 1 and 2 for objective evaluation, and in Appendix A.5 for subjective evaluation. We also conduct two experiments to evaluate dataset scalability using 0.5K, 5.5K, 10.5K, and 50.5K-hour subsets. In the first experiment, we examine the performance of text-speech alignment based on data size for the same target latent. In the second experiment, following NaturalSpeech 3, we

train both the Mel-VAE and DiTTo on the same subsets. The training details and results are in Appendix A.13, showing that model performance improves as the data scale increases.

## 5.2 KEY ASPECTS BEHIND THE SCENE (**RQ2**)

Table 4: Performance of the English-only *cross-sentence* task shows that DiT fits better than U-Net. We use ground truth length during sampling, set the diffusion steps to 250 for WER and SIM-r, and 25 for Inference Time. The noise schedule scale-shift is set to 0.3, and the classifier-free guidance scale to 5.0. The boldface indicates the best result.

| Model | WER ↓ | SIM-r ↑ | Inference Time ↓ |
|---|---|---|---|
| *U-Net* | 3.70 | 0.3890 | 1.328s |
| *Flat-U-Net* | 2.97 | 0.5471 | 1.310s |
| DiTTo-*mls* | **2.93** | **0.5877** | **0.903s** |

**DiT Fits Better Than U-Net**  We experimentally demonstrate that the DiT (Peebles & Xie, 2023) architecture is more suitable for TTS than the commonly used U-Net (Ronneberger et al., 2015), especially when domain-specific factors are eliminated (Lovelace et al., 2024; Gao et al., 2023). We implement three DiTTo variants (1) using U-Audio Transformer (U-AT) proposed by (Lovelace et al., 2024) (referred to as *U-Net*), (2) then eliminating the down/up sampling of U-AT (referred to as *Flat-U-Net*), and (3) with the same DiTTo-en architecture (referred to as DiTTo-*mls*). The results are summarized in Table 4. Recall that we use the Mel-VAE latent, which is already compressed about seven times more than EnCodec (Défossez et al., 2023). This additional compression leads to significant information loss and poor output quality in *U-Net*. In *Flat-U-Net*, it proves to be a more suitable choice than U-Net but still leaves room for improvement. We further conduct ablation studies on our model architecture in Section 6.

**Variable Length Modeling Outperforms Fixed Length Modeling**  We compare and analyze four different approaches to speech length modeling: (1) *Fixed-length*, trained with a maximum of 20 seconds of speech where the target latent includes variable-length paddings; (2) *Fixed-length-full*, similar to *Fixed-length* but predicting all paddings within the maximum length; (3) *SLP-CE*, DiTTo-*mls* with our proposed speech length predictor using top-$k$ sampling ($K = 20$); and (4) *SLP-Regression*, similar to *SLP-CE* but with a regression objective. The further details are summarized in Appendix A.14. Table 5 shows that in fixed-length modeling, performance degrades as more padding is added. It also demonstrates that variable-length modeling significantly outperforms all fixed-length models by a wide margin. Although the CE-based and regression-based speech length predictors present a trade-off between WER and SIM, we employ CE loss to enable sampling of the speech length during inference. The variable-length modeling also enables speech rate control by changing the total length of the generated speech latent, as illustrated in Figure 2.

Table 5: Speech length modelings. SLP-CE uses top-$k$ sampling with $K = 20$.

| Model | WER ↓ | SIM-r ↑ | Inference Time ↓ |
|---|---|---|---|
| *fixed-length-full* | 8.89 | 0.4078 | 1.254s |
| *fixed-length* | 6.81 | 0.4385 | 1.265s |
| *SLP-CE* | 5.58 | **0.4961** | 0.948s |
| *SLP-Regression* | **5.36** | 0.4636 | **0.930s** |

**Aligned Text-Speech Embeddings Improve Performances**  DiTTo uses cross-attention for text conditioning, which can be influenced by the distance between text and speech representations. To validate the effect of aligned text-speech embeddings, we train DiTTo-*mls* variants with two different text encoders: 1) ByT5 (Xue et al., 2022) (we use ByT5-base), which is trained solely on a large corpus of multilingual text, and 2) SpeechT5 (Ao et al., 2022), which is trained jointly on English-only text and corresponding speech. In Table 6, we observe that DiTTo-*mls* trained with SpeechT5 (in the 2nd row) outperforms the model trained with ByT5-base (in the 1st row) in terms of speech accuracy. Given that the encoder size for SpeechT5 is 85M and for ByT5-base is 415M, and that ByT5 trains on a significantly larger dataset, this indicates that aligning text embeddings with speech embeddings effectively enhances TTS performance, independent of the model or data

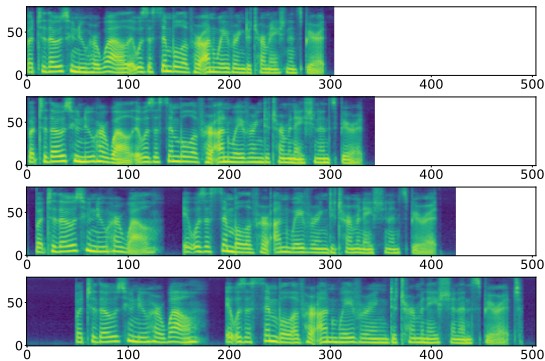

Figure 2: **Speech rate controllability** is illustrated through mel-spectrograms of generated speech at various rates. The speech rate decreases from top to bottom, achieved by adjusting the latent length from 38 to 63. The transcript is *"But to those who knew her well, it was a symbol of her unwavering determination and spirit."* The $x$-axis represents the length, and the $y$-axis represents the channels of the mel-spectrogram. Speech samples are available at `https://ditto-tts.github.io`.

Table 6: Performances of English-only *cross-sentence* task for Mel-VAE++. $U$ (short for 'Unimodal') refers to training on either text or speech alone, while $J$ (short for 'Jointly-trained') refers to modeling both text and speech together. The boldface indicates the best result.

| Text Encoder | Neural Audio Codec | WER ↓ | CER ↓ | SIM-o ↑ | SIM-r ↑ |
|---|---|---|---|---|---|
| $U$ (ByT5-base) | $U$ (Mel-VAE) | 6.22 | 3.82 | **0.5482** | 0.5945 |
| $J$ (SpeechT5) | $U$ (Mel-VAE) | 3.07 | 1.15 | 0.5423 | 0.5858 |
| $U$ (ByT5-base) | $J$ (Mel-VAE++) | 3.11 | 1.17 | 0.5323 | 0.5965 |
| $J$ (SpeechT5) | $J$ (Mel-VAE++) | **2.99** | **1.06** | 0.5364 | **0.5982** |

size. Therefore, it is advisable to select a pre-trained text encoder that has been jointly modeled with speech when choosing among various off-the-shelf options.

Building on this insight, we fine-tune Mel-VAE on the same MLS dataset for 200K steps by setting $\lambda > 0$ in Eq. (2). This process results in two fine-tuned models, referred to as Mel-VAE++, each aligned with ByT5 (in the 3rd row) and SpeechT5 (in the 4th row), respectively. What we additionally find is the followings: (1) The pair of ByT5 and Mel-VAE++ (in the 3rd row) performs significantly better than the pair of ByT5 and Mel-VAE (in the 1st row). (2) This performance is similar to that of the pair of SpeechT5 and Mel-VAE (in the 2nd row), indicating that having at least one encoder jointly trained contributes greatly to performance improvement. (3) While the best performance is observed when both are jointly trained (in the last row), the degree of improvement is much smaller compared to the improvement from (1) or (2). Aggregating all results, we can conclude that the closer the text-speech representation, the greater the improvement in DiTTo's text-speech alignment quality. Notably, fine-tuning only the neural audio codec, without fine-tuning the text encoder, is efficient to enhance alignment and subsequently improve TTS performance. This approach may offer a significant advantage since fine-tuning the language model is more time-consuming. In Appendix A.15, we attempt to compare various semantic injection methods as well.

Unless otherwise specified, all models follow the training setup described in Appendix A.16.

## 6 ABLATION STUDY

**Model Architecture Details** As listed in Table 7, we consider several model architecture options, and our final design demonstrates optimal performance. Starting from *Flat-U-Net* in Section 5.2, we remove the residual and convolution block, making our model resemble the DiT architecture. As implemented in Pixart-alpha (Chen et al., 2023a), we then modify AdaLN to be shared (DiTTo-*local-adaln*). This reduces model parameters by 30.5% (for XL models) and surprisingly improves performance as well. We add a long skip connection to DiT inspired by U-Net. Notably, connecting the hidden layers inside the transformer block as in U-ViT (Bao et al., 2023) (DiTTo-*uvit-skip*) or without any skip connection (DiTTo-*no-skip*) is less effective in terms of model convergence than

Table 7: Performances of English-only cross-sentence task for ablation study. All models use SpeechT5 as text encoder and original Mel-VAE as audio latent codec, and comsume ground truth length during sampling. We set the diffusion steps to 250 for WER and SIMs and 25 for Inference Time, noise schedule scale-shift to 0.3, and classifier-free guidance scale to 5.0. Results confirm the effectiveness of the architectural design of our model. The boldface indicates the best result.

| Model | WER ↓ | SIM-o ↑ | SIM-r ↑ | Inference Time ↓ | Codec PESQ ↑ | Codec ViSQOL ↑ |
|---|---|---|---|---|---|---|
| DiTTo-*local-adaln* | 3.38 | 0.5263 | 0.5673 | 0.937s | | |
| DiTTo-*uvit-skip* | 3.17 | 0.5456 | 0.5848 | 0.940s | | |
| DiTTo-*no-skip* | 3.30 | 0.5304 | 0.5727 | 0.905s | 2.95 | 4.66 |
| DiTTo-*no-pooled-text* | 3.00 | 0.5410 | 0.5791 | 0.912s | | |
| DiTTo-*no-rvq-decoding* | 2.97 | 0.5468 | **0.5883** | **0.894s** | | |
| DiTTo-*mls* (from Table 4) | **2.93** | 0.5467 | 0.5877 | 0.903s | | |
| DiTTo-*encodec* | 4.19 | 0.5105 | 0.5460 | n/a | 2.59 | 4.26 |
| DiTTo-*dac-24k* | 7.21 | **0.5478** | 0.5545 | n/a | **4.37** | **4.91** |
| DiTTo-*dac-44k* | 14.58 | 0.5391 | 0.5597 | n/a | 3.74 | 4.85 |

a residual connection that links before and after the transformer block (we now arrive at our final architecture, DiTTo-*mls*). Additionally, we adopt the pooled-text method from GenTron (Chen et al., 2023b), which borrows from class embeddings in image domain. We verify this by training DiTTo without pooled-text (DiTTo-*no-pooled-text*) and observe a slight decline in performance. To assess the impact of quantization, we decode DiTTo output without quantization (DiTTo-*no-rvq-decoding*) and observe minimal performance differences (additional discussion is provided in Appendix A.17). Furthermore, we examine the effect of training steps on performance in Appendix A.18.

**Why Mel-VAE? Comparisons with EnCodec and DAC**   In this section, we show that Mel-VAE is the most suitable target latent by comparing its performance with commonly used audio codecs, EnCodec (Défossez et al., 2023) and DAC (Kumar et al., 2024). We implement variants of DiTTo-*mls*, naming them DiTTo-*encodec*, *dac-24k*, and *dac-44k*, where the two DAC versions correspond to different target audio sample rates. The results in Table 7 show that our model performs best in both WER and SIM-r (we discuss SIM-o in Appendix A.19). We cannot measure total inference time due to the absence of a speech length predictor, but their 7-8 times longer latents significantly slow generation, even with ground truth lengths. Training with DAC is challenging due to its latent dimension of 1024—twice that of Mel-VAE—and its 7-8 times higher latent code rate. While DAC scores higher on PESQ (Rix et al., 2001) and ViSQOL (Chinen et al., 2020) metrics (see Appendix A.3 for measurement details), Mel-VAE remains more suitable for training DiTTo due to its shorter latent length and lower dimensionality, allowing for more efficient training and inference. EnCodec, despite having a smaller latent dimension of 128, also suffers from a longer latent and poorer codec performance, which negatively affects the final performance. Although codec quality and latent dimension impact outcomes, the slower generation and training challenges with longer latent lengths suggest it may be a critical factor. To verify this, we conduct additional experiments on Mel-VAE variants with different compression ratios (2×, 4×, and 8× as in the original Mel-VAE). The results in Appendix A.20 confirm that performance improves as the latent length decreases. We also discuss using mel-spectrograms as targets without latent representations in Appendix A.21.

## 7   CONCLUSION

In this work, we introduce DiTTo-TTS, a latent diffusion model (LDM) for text-to-speech (TTS) that compares to or surpasses existing state-of-the-art models in zero-shot naturalness, intelligibility, and speaker similarity—all without relying on domain-specific elements like phonemes and durations. This achievement is made possible through rigorous model architecture search, variable-length modeling facilitated by total length prediction, and discovering that aligning text and speech latents via joint modeling is crucial for effective cross-attention learning. DiTTo-TTS also scales well with both data and model sizes. Thanks to DiTTo-TTS breaking down the barriers between TTS and other fields, we can now approach TTS in a similar manner. For future work, we plan to establish DiTTo as a baseline upon which advancements from cutting-edge LDM research across various modalities can be applied. Additionally, leveraging cross-domain knowledge from other pre-trained DiT models appears promising. Finally, we aim to enable DiTTo-TTS to understand natural language instructions, making it more suitable for interactive tasks.

## 8   ETHICS STATEMENTS

DiTTo-TTS is a zero-shot text-to-speech model that leverages latent diffusion to enable efficient large-scale learning and inference without relying on domain-specific elements such as phonemes or durations. By generating natural and intelligible speech across a wide range of voices with minimal input, DiTTo-TTS provides significant advantages in scalability and flexibility. However, its ability to synthesize any voice—including those of real individuals—with minimal data input presents potential risks, particularly in terms of misuse for malicious purposes such as voice spoofing, impersonation, or deepfake audio generation. These risks highlight the importance of ethical considerations and responsible deployment of such powerful technology.

Given the growing concerns around synthetic media and its societal impact, it becomes crucial to address these issues proactively. Developing robust detection systems capable of accurately identifying synthetic audio generated by models like DiTTo-TTS is a necessary step to mitigate these risks. Additionally, strict protocols must be established to assess the model's impact, including guidelines for responsible use, consent in voice cloning, and regulatory frameworks to prevent malicious applications. These measures will ensure that the benefits of DiTTo-TTS can be realized while minimizing its potential for harm.

In light of these ethical considerations, we use only publicly available speech datasets for training and evaluation, ensuring they contain no personally identifiable information (PII) and are legally permissible. We rely exclusively on transcripts and audio clips, explicitly excluding any metadata or speaker IDs to maintain anonymity. Consequently, the model is trained solely on pairs of text and corresponding audio. Furthermore, these datasets are distributed under licenses permitting research use—such as MLS (CC BY 4.0), GigaSpeech (non-commercial research under its Terms of Access), LibriTTS-R (CC BY 4.0), VCTK (Open Data Commons Attribution License (ODC-By) v1.0), LJSpeech (Public Domain), Expresso Dataset (CC BY-NC 4.0), and AI-Hub datasets (including AIHub 14, AIHub 15, KsponSpeech, AIHub-broadcast, and AIHub-expressive, as noted in our Acknowledgment section in accordance with policy)—all of which allow research and, in most cases, commercial usage. We ensure that our data usage aligns with ethical and legal requirements.

## 9   REPRODUCIBILITY STATEMENTS

We present Figure 1, which provides a visual overview of the DiTTo-TTS model architecture, and we describe the detailed components of the architecture in Section 3.3. The corresponding model hyperparameters are discussed in Appendix A.5 to give further insight into the configurations used. To ensure the reproducibility of our experiments, we supply comprehensive details across several sections. Appendix A.2 contains the complete list and statistics of the training data used, while Section 4 covers the data preprocessing procedures. The training configurations and evaluation methodologies are provided in both Section 4 and Appendix A.3, ensuring all experimental parameters are transparent. Additionally, if legal concerns can be addressed, we plan to gradually release the inference code, pre-trained weights, and eventually the full training implementation, enabling the research community to further explore and validate our findings.

### ACKNOWLEDGMENT

The authors would like to express our gratitude to Kangwook Lee for the valuable discussions. We also extend our thanks to Minki Kang and Minkyu Kim for their thorough proofreading of the paper, and to Beomsoo Kim and Gibum Seo for their essential support in data handling and verification.

This research (paper) used datasets from 'The Open AI Dataset Project (AI-Hub, S. Korea)'. All data information can be accessed through 'AI-Hub (www.aihub.or.kr)'

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

## A APPENDIX

### A.1 RELATED WORK

**Discussion on Seed-TTS** Seed-TTS (Anastassiou et al., 2024), particularly its variant Seed-TTS$_{\text{DiT}}$, demonstrates that fully diffusion-based models achieve superior performance without relying on domain-specific factors such as phonemes or phoneme-level durations. By providing only the total duration for the diffusion model, Seed-TTS$_{\text{DiT}}$ removes the need for precise phoneme durations, enabling the DiT architecture to flexibly learn fine-grained durations. This approach illustrates that high-quality speech synthesis is achievable without detailed phoneme-level information. While Seed-TTS highlights the potential of simple modeling approaches in TTS, it does not offer a comprehensive comparison with diverse baselines and explore the critical factors enabling LDMs to succeed without domain-specific factors—both of which are the central focus of our study. We specifically address two key research questions, as discussed in Section 5: *RQ1*—Can LDMs achieve state-of-the-art performance in text-to-speech tasks at scale without relying on domain-specific factors, similar to their successes in other domains? *RQ2*—If so, what are the primary factors that enable their success? Our study systematically addresses these questions through extensive empirical investigations. We demonstrate that LDM-based TTS, even without domain-specific factors, achieves performance that is superior or comparable to state-of-the-art autoregressive (AR) and non-autoregressive (Non-AR) TTS baselines, enabling a fair evaluation of its capabilities within the broader research landscape. Furthermore, we identify the key factors driving this success and provide practical insights into a new modeling paradigm for TTS systems. Notable contributions include methodological advancements such as the semantic injection of Mel-VAE++, a speech length predictor, and architectural ablations involving a long skip connection, offering actionable guidance for optimizing the promising modeling paradigm.

**Large-scale TTS** Recently, large-scale TTS research progresses actively in two main directions: LLM-based autoregressive (AR) TTS and non-autoregressive (Non-AR) TTS. A prominent feature of LLMs is the scalability (Shoeybi et al., 2019; Kaplan et al., 2020) and their proficiency in zero-shot learning tasks, demonstrating significant capabilities without prior specific training on those tasks (Brown et al., 2020; Touvron et al., 2023; OpenAI, 2022; 2023). Efforts to replicate LLM's capability in different modalities have shown progress, including vision (Liu et al., 2024; Li et al., 2023; Fathullah et al., 2024) and audio (Wang et al., 2023; Borsos et al., 2023; Kharitonov et al., 2023; Chu et al., 2023). VALL-E (Wang et al., 2023) employs EnCodec (Défossez et al., 2023) for speech-to-token mapping, posing TTS tasks as AR language modeling tasks, thus enabling zero-shot capabilities in the speech domain. CLaM-TTS (Kim et al., 2024a) introduces Mel-VAE to achieve superior token length compression, and enables a language model to predict multiple tokens simultaneously. Although this approach removes the need for cascaded modeling (Wang et al., 2023) to manage the number of token streams, their resource-intensive inference processes limit their applications (Ainslie et al., 2023). On the other hand, Non-AR generative models are employed to enhance the efficiency of TTS systems. Voicebox (Le et al., 2023) utilizes a flow matching (Lipman et al., 2022) to generate speech, effectively casting the TTS task into a speech infilling task. NaturalSpeech series (Shen et al., 2024; Ju et al., 2024), building upon recent advances in the Latent Diffusion Model (LDM) (Rombach et al., 2022), incorporate auxiliary modules for controllability of various speech attribute such as content, prosody, and timbre. However, requiring supplementary data beyond speech-transcription pairs, and auxiliary modules hinder efficiency and scalability.

**Neural Audio Codec** Neural audio codecs, which effectively compress various types of audio using neural networks, are used as part of many TTS systems (Shen et al., 2024; Kim et al., 2024a; Wang et al., 2023). Recent advancements employ an encoder-decoder architecture coupled with Residual Vector Quantization (RVQ) (Gray, 1984; Vasuki & Vanathi, 2006; Lee et al., 2022) to transform raw audio waves into discretized tokens. For example, EnCodec (Défossez et al., 2023) converts 24,000 Hz mono waveforms into 75 Hz latents. With a similar architecture, by focusing the compression specifically on speech rather than general audio signals, Mel-VAE (Kim et al., 2024a) achieves approximately 10.76 Hz latents by compressing the mel-spectrogram. This reduction significantly lowers the computational cost of the speech generation module. Another research direction of improving neural audio codecs for TTS systems is injecting semantic information using large language models (LLMs) (Zhang et al., 2024).

## A.2 DATASET DETAILS

**English** We use the following datasets: (1) Multilingual LibriSpeech (MLS) (Pratap et al., 2020), which comprises transcribed speech from multiple speakers and languages, sourced from LibriVox audiobooks. (2) GigaSpeech (Chen et al., 2021), containing multi-domain speeches, such as audiobooks, podcasts, and YouTube videos, with human transcriptions. This dataset includes audio from multiple speakers but lacks speaker information. (3) LibriTTS-R (Koizumi et al., 2023), a restored version of the LibriTTS (Zen et al., 2019) corpus, sharing the same metadata. (4) VCTK (Veaux et al., 2016) and (5) LJSpeech (Ito & Johnson, 2017), which are widely used English datasets in the speech synthesis community, with VCTK being multi-speaker and LJSpeech being single-speaker. Additionally, we include the (6) Expresso Dataset (Nguyen et al., 2023) in DiTTo-en training. This high-quality (48,000 Hz) expressive speech collection features read speech and improvised dialogues from four speakers, totaling 40 hours.

We train a speech length predictor for DiTTo-en using the same datasets, but also include the LibriSpeech (Panayotov et al., 2015) dataset. This is due to the dependency on text normalization. MLS datasets consist of audio recordings, each ranging from about 10 to 20 seconds in length, paired with normalized text. It affects the speech length predictor, making it output lengths based on whether the input text is normalized. For example, if we input normalized text, the model tends to predict its speech length in the same range as MLS. While the text in LibriSpeech is also normalized, it offers more varied speech lengths, thus reducing the dependency on text normalization when determining speech lengths.

**Korean** We use the following datasets: (1) AIHub 14[3], which features recordings of everyday people reading provided script sentences. (2) AIHub 15[4], which has recordings of 50 professional voice actors expressing seven emotions (joy, surprised, sad, angry, scared, hate, neutral), five speaking styles (narrating, reading, news-like, dialogic, broadcasting), and three vocal ages (kid, young, old). (3) KsponSpeech (Bang et al., 2020), which consists of 2,000 speakers, each recording individual free speech on various topics in a quiet environment. The transcription follows specific guidelines regarding laughter, breathing, and more. (4) AIHub-broadcast[5], which is designed for speech recognition and includes 10,000 hours of multi-speaker conversations recorded at 16,000 Hz, covering a wide range of 22 categories without annotations for speaker identity or emotional state. We also include 5) AIHub-expressive[6], a dataset designed for speech synthesis, comprising 1,000 hours of speech recorded at 44,100 Hz. This dataset features 89 speakers and includes 7 speech styles (monologue, dialogue, storytelling, broadcast, friendly, animation, and reading) and 4 emotional tones (happiness, sadness, anger, and neutral).

**Other Language** We use subsets of MLS including German, Dutch, French, Spanish, Italian, Portuguese, and Polish.

**Dataset Preprocessing** Audio stream and its metadata are preprocessed and stored into parquet format. This allows fetching data with minimal IO-overhead from Network Attached Storage, allowing faster training. Parquets consisting of 10K pairs of audio and its metadata are compressed into TAR format, which further compresses to minimize data read overhead. While training, TAR files are decompressed, and parquets are deserialized into audio streams and its corresponding metadata in json format.

## A.3 EXPERIMENTAL SETUP DETAILS

For SIM, we borrow SIM-o and SIM-r from Voicebox (Le et al., 2023), where SIM-o measures the similarity between the generated speech and the original target speech, while SIM-r measures the similarity between the target speech reconstructed from the original speech using pre-trained

---

[3]https://www.aihub.or.kr/aihubdata/data/view.do?currMenu=115&topMenu=100&aihubDataSe=realm&dataSetSn=542

[4]https://www.aihub.or.kr/aihubdata/data/view.do?currMenu=115&topMenu=100&aihubDataSe=realm&dataSetSn=466

[5]https://www.aihub.or.kr/aihubdata/data/view.do?currMenu=115&topMenu=100&aihubDataSe=realm&dataSetSn=463

[6]https://www.aihub.or.kr/aihubdata/data/view.do?currMenu=115&topMenu=100&aihubDataSe=realm&dataSetSn=71349

autoencoder and vocoder. For English-only evaluations, we transcript generated audio using the CTC-based HuBERT-Large model[7] (Hsu et al., 2021). For Multilingual Evaluations, we utilize OpenAI's Whisper model[8] (Radford et al., 2023). Text normalization is conducted using NVIDIA's NeMo-text-processing[9] (Zhang et al., 2021; Bakhturina et al., 2022). In both evaluations, WavLM-TDCNN[10] (Chen et al., 2022) is employed to evaluate SIM-o and SIM-r (Le et al., 2023). If required, we also measure PESQ (Rix et al., 2001) and ViSQOL (Chinen et al., 2020) for some codecs using a fixed test set comprised of 100 randomly sampled instances from each dataset listed in Table 6 of CLaM-TTS (Kim et al., 2024a).

All objective and subjective evaluation samples were downsampled to 16,000 Hz, except for subjective evaluation on demo samples. We conduct subjective evaluations using Amazon Mechanical Turk (MTurk) with US-based evaluators. For SMOS, evaluators assess the likeness of samples to the provided speech prompts, considering speaker similarity, style, acoustics, and background disturbances. In CMOS, evaluators compare the overall quality of a synthesized sample to a reference. They use a given scale to judge whether the synthesized version is superior or inferior to the reference. SMOS employs a 1 to 5 integer scale, where 5 signifies top quality. CMOS uses a scale from -3 (indicating the synthesized speech is much worse than the reference) to 3 (indicating it's much better), with 1-unit intervals. Samples receive 6 and 12 ratings for SMOS and CMOS, respectively. Figure 3 presents the instructions given to the evaluators for the SMOS study, while Figure 4 presents the instructions for the CMOS study.

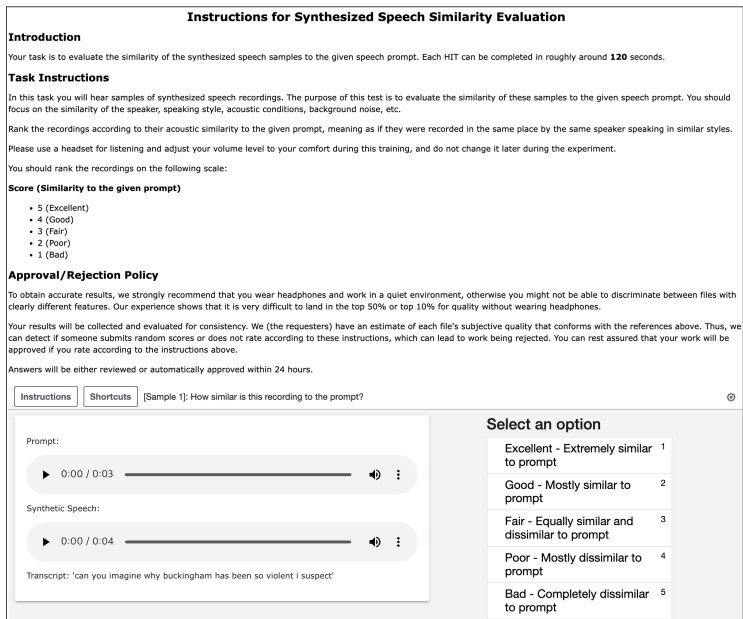

Figure 3: Similarity mean opinion score (SMOS) instruction.

### A.4 BASELINE DETAILS

We include StyleTTS 2 (Li et al., 2024) and XTTS-v2 (Casanova et al., 2024) as open-source TTS models in our objective evaluation. For the RAVDESS (Livingstone & Russo, 2018) evaluation described in Section 5.1, we use demo samples provided by NaturalSpeech 3 (Ju et al., 2024), which include the following additional models: Mega-TTS 2 (Jiang et al., 2024), StyleTTS 2 (Li et al., 2024), and HierSpeech++ (Lee et al., 2023a). Additionally, we include SimpleSpeech (Yang et al., 2024b) and E2 TTS (Eskimez et al., 2024) as concurrent works. We use open-sourced checkpoints

---

[7] https://huggingface.co/facebook/hubert-large-ls960-ft

[8] https://github.com/openai/whisper/blob/main/model-card.md: "large-v2"

[9] https://github.com/NVIDIA/NeMo-text-processing

[10] https://github.com/microsoft/UniSpeech/tree/main/downstreams/speaker_verification

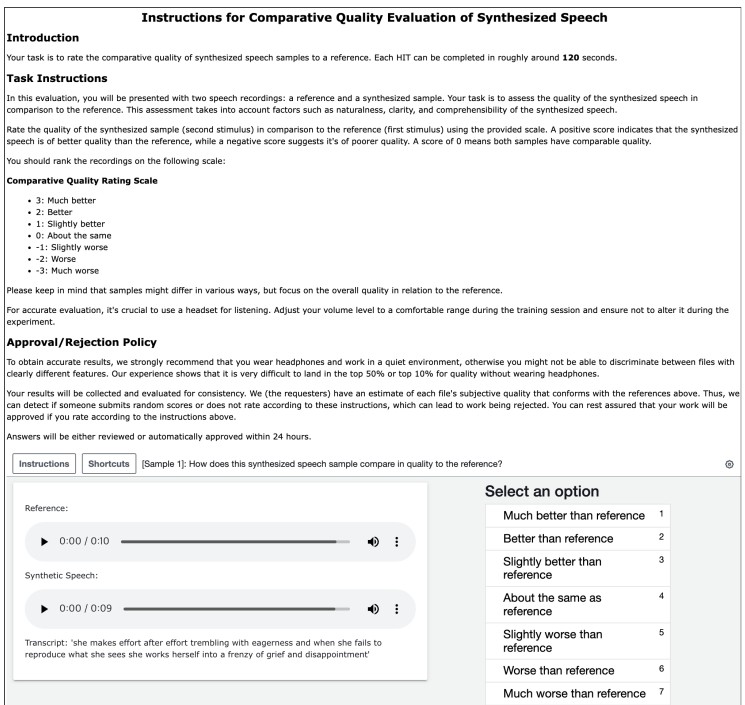

Figure 4: Comparative mean opinion score (CMOS) instruction.

of YourTTS [11], Simple-TTS [12], StyleTTS 2 [13], XTTS-v2 [14], and the official checkpoint of CLaM-TTS for evaluations. When conducting subjective evaluations, all samples are resampled to 16,000 Hz. The followings are the details of the baselines we used to compare with our model:

- VALL-E (Wang et al., 2023): A zero-shot TTS model integrates autoregressive and non-autoregressive models. We show the objective metric scores directly extracted from the paper. We also present subjective evaluation results by comparing our samples with demo samples, which are at sample rates of 22,050 Hz and 16,000 Hz, respectively.

- SPEAR-TTS (Kharitonov et al., 2023): A multi-speaker TTS model that treats TTS as a two-step sequence-to-sequence process. It first converts text into high-level semantic tokens, similar to "reading," and then transforms these semantic tokens into low-level acoustic tokens, akin to "speaking." We present the objective metric scores directly from the paper.

- CLaM-TTS (Kim et al., 2024a): An advanced neural codec language model that uses probabilistic residual vector quantization to compress token length, enabling the simultaneous generation of multiple tokens, in contrast to VALL-E's cascaded modeling approach. We offer both objective and subjective comparisons between our samples and those generated from the official model checkpoints.

- YourTTS (Casanova et al., 2022): A conventional TTS model based on the VITS (Kim et al., 2021) framework for zero-shot multi-speaker and multilingual training. We show both objective and subjective comparisons between our samples and those generated from the model checkpoints. We use the official checkpoint [15].

- Voicebox (Le et al., 2023): A generative model employs a non-autoregressive flow-based architecture to tackle TTS as a text-guided speech infilling task, utilizing a large dataset.

---

[11] https://github.com/Edresson/YourTTS
[12] https://github.com/asappresearch/simple-tts
[13] https://github.com/yl4579/StyleTTS2
[14] https://huggingface.co/coqui/XTTS-v2
[15] https://github.com/Edresson/YourTTS

We present the objective metric scores directly from the paper. Additionally, we provide subjective evaluation results by comparing our samples with demo samples, recorded at sample rates of 22,050 Hz and 16,000 Hz, respectively.

- Simple-TTS (Lovelace et al., 2024): A Latent Diffusion Model (LDM) based on U-ViT (Bao et al., 2023) serves as an alternative to complex TTS synthesis pipelines, removing the need for phonemizers, forced aligners, or detailed multi-stage processes. We provide both objective and subjective comparisons between our samples and those generated from the model checkpoints. We use the official checkpoint [16].

- NaturalSpeech 2 (Shen et al., 2024): A non-autoregressive TTS system that incorporates a neural audio codec employing residual vector quantizers to obtain quantized latent vectors. These vectors serve as input for a diffusion model, which generates continuous latent vectors conditioned on text input. Additionally, the system employs a speech prompting mechanism along with duration and pitch predictors. We present subjective evaluation results by comparing our samples with demo samples, which are at sample rates of 22,050 Hz and 16,000 Hz, respectively.

- NaturalSpeech 3 (Ju et al., 2024): A non-autoregressive TTS system with factorized diffusion models based on a factorized vector quantization (FVQ) to disentangle speech waveforms into distinct subspaces such as content, prosody, timbre, and acoustic details. We present subjective evaluation results by comparing our samples with demo samples, which are at sample rates of 22,050 Hz and 16,000 Hz, respectively.

- Mega-TTS (Jiang et al., 2023): A TTS system that decomposes speech into attributes like content, timbre, prosody, and phase, with each attribute modeled by a module using appropriate inductive biases. We show subjective evaluation results by comparing our samples with demo samples, which have sample rates of 22,050 Hz and 16,000 Hz, respectively.

- E3 TTS (Gao et al., 2023): A simple and efficient diffusion based TTS model which takes plain text as input. Unlike previous work, it doesn't rely on intermediary representations like spectrogram features or alignment data. We present subjective evaluation results by comparing our samples with demo samples, which are at sample rates of 22,050 Hz and 24,000 Hz, respectively.

- Mega-TTS 2 (Jiang et al., 2024): A TTS model that introduces a prompting mechanism for zero-shot TTS that separates prosody and timbre encoding, enabling high-quality, identity-preserving speech synthesis with enhanced transferability and control. We present subjective evaluation results using demo samples at 16,000 Hz, following the approach of NaturalSpeech 3 (Ju et al., 2024).

- StyleTTS 2 (Li et al., 2024): A TTS model that uses style diffusion and adversarial training with large speech language models, without requiring reference speech. We present subjective evaluation results using demo samples at 16,000 Hz, following the approach of NaturalSpeech 3 (Ju et al., 2024). For objective evaluation, we use samples generated from the model checkpoints. We use the official checkpoint [17].

- HierSpeech++ (Lee et al., 2023a): A fast and robust zero-shot speech synthesizer for both text-to-speech and voice conversion, utilizing a hierarchical speech synthesis framework. We present subjective evaluation results using demo samples at 16,000 Hz, following the approach of NaturalSpeech 3 (Ju et al., 2024).

- XTTS-v2 (Lee et al., 2023a): A TTS system that addresses the limitations of existing Zero-shot Multi-speaker TTS models by enabling multilingual training and faster, more efficient voice cloning across 16 languages. For objective evaluation, we use samples generated from the model checkpoints. We use the official checkpoint [18].

- SimpleSpeech (Gao et al., 2023): A non-autoregressive TTS system based on diffusion that operates without alignment information and generates speech from plain text. It uses a novel speech codec model (SQ-Codec) with scalar quantization to map speech into a finite and compact latent space, simplifying diffusion modeling. We present subjective evaluation results using demo samples at 16,000 Hz.

---

[16] https://github.com/asappresearch/simple-tts
[17] https://github.com/yl4579/StyleTTS2
[18] https://huggingface.co/coqui/XTTS-v2

- E2 TTS (Gao et al., 2023): A non-autoregressive TTS method that converts text into a character sequence with filler tokens and trains a flow-matching-based mel spectrogram generator using an audio infilling task. It does not require additional components or complex alignment techniques, offering a simplified approach to TTS. We present subjective evaluation results using demo samples at 16,000 Hz.

## A.5 MEL-VAE AND MODEL CONFIGURATION

Table 8: $d_c$ is the embedding dimension of pretrained encoder, and $d_h$ is the hidden dimension of DiTTo. SiLU is used as activation function between two linear layers.

| Configuration | Num Layers | Input Dim | Hidden Dim | Out Dim |
|---|---|---|---|---|
| Timestep Embedder | 2 | 256 | $d_h$ | $d_h$ |
| Text Embedder | 2 | $d_c$ | $d_h$ | $d_h$ |
| Skip Block | 2 | $2 \times d_h$ | $d_h$ | $d_h$ |
| Final MLP | 2 | $d_c$ | $d_h$ | $d_h$ |
| AdaLN Modulation | 1 | $d_h$ | - | $9 \times d_h$ |

**Mel-VAE**: We utilized the pretrained Mel-VAE, which was obtained through collaboration with the authors of CLaM-TTS (Kim et al., 2024a). Our preference for Mel-VAE over neural audio codes such as EnCodec (Défossez et al., 2023) is based on the observation that latent audio representations closely related to text yield better performance in TTS. We illustrate the autoencoding process of Mel-VAE as follows: The encoder maps a mel-spectrogram into a sequence of latent representations $z_{1:T}$, which are then processed by a residual vector quantizer $RQ_\psi(\cdot)$. This quantizer converts each latent vector $z_t$ at time $t$ into a quantized embedding $\hat{z}_t = \sum_{d=1}^{D} e_\psi(c_{t,d}; d)$. Each embedding at depth $d$, $e_\psi(c_{t,d}; d)$, is selected to minimize $\|(z - \sum_{d'=1}^{d-1} e_\psi(c_{t,d'}; d')) - e_\psi(c_{t,d}; d)\|^2$ among the codebook embeddings $\{e_\psi(c', d)\}_{c'=1}^{V}$, where the codebook size is $V$. The decoder then reconstructs the mel-spectrogram from the sequence of quantized latent representations $\hat{z}_{1:T}$.

**Model Configuration**: Our architecture is based on DiT (Peebles & Xie, 2023) with several modifications. We extract text embeddings from a pre-trained text encoder, either from speechT5 (Ao et al., 2022) or ByT5 (Xue et al., 2022), and use these embeddings as hidden states in cross-attention layers. Adaptive layer normalization (AdaLN) layers condition on both time and text embeddings. AdaLN outputs modulate the layers within each transformer block. We apply RoPE (Su et al., 2024) in self-attention layers. Following (Chen et al., 2023a), we use a single global-AdaLN, sharing AdaLN parameters across all layers instead of using separate AdaLN for each layer. The global AdaLN layer conditions on both time and text, which are embedded and transformed with MLP layers. AdaLN generates scaling, shifting, and gating vectors for each self-attention, cross-attention, and MLP layer. Gating vectors modulate the outputs of each layer over the skip connections in transformer blocks. The vectors generated by AdaLN are further shifted with layer-independent learnable vectors. The MLP block in each transformer block adopts the gating mechanism from (Team et al., 2024), which is used in the first linear layer of the MLP block. Inspired by (Bao et al., 2023), we insert a long skip connection from the input to the output of the last transformer block, differing from (Bao et al., 2023), where U-Net-like (Ronneberger et al., 2015) long skip connections are used between transformer blocks at opposing ends. The detailed configurations are shown in Table 8, and the parameters for the four different model sizes are listed in Table 9. Additionally, Table 10 summarizes the subjective evaluation results for each model size. As the model size increases, the

Table 9: Model parameters across different versions of DiTTo

| Configuration | **DiTTo-S** | **DiTTo-B** | **DiTTo-L** | **DiTTo-XL** |
|---|---|---|---|---|
| Num Layers | 12 | 12 | 24 | 28 |
| Hidden Dim ($d_h$) | 384 | 768 | 1024 | 1152 |
| Num Heads | 6 | 12 | 16 | 16 |

Table 10: Human evaluations on *cross-sentence* task with 40 LibriSpeech test-clean speakers along with four different model scales (S, B, L, and XL). The value for DiTTo-en-XL is taken from Table 3. SMOS scores include a 95% confidence interval. The boldface indicates the best result among models and *recon* indicates the reconstructed audio by Mel-VAE followed by vocoder.

| Model | DiTTo-en-S | | DiTTo-en-B | | DiTTo-en-L | | DiTTo-en-XL | |
|---|---|---|---|---|---|---|---|---|
| | SMOS | CMOS | SMOS | CMOS | SMOS | CMOS | SMOS | CMOS |
| Simple-TTS | - | -1.55 | - | -1.90 | - | -1.81 | 2.15±0.19 | -1.64 |
| CLaM-en | - | -0.05 | - | -0.45 | - | -0.44 | 3.42±0.16 | -0.52 |
| DiTTo-en-en | **3.45**±**0.13** | **0.00** | **3.55**±**0.14** | **0.00** | **3.66**±**0.13** | **0.00** | **3.91**±**0.16** | **0.00** |
| Ground Truth (*recon*) | - | +0.73 | - | +0.26 | - | +0.16 | 4.07±0.14 | +0.11 |
| Ground Truth | - | +0.81 | - | +0.49 | - | +0.38 | 4.08±0.14 | +0.13 |

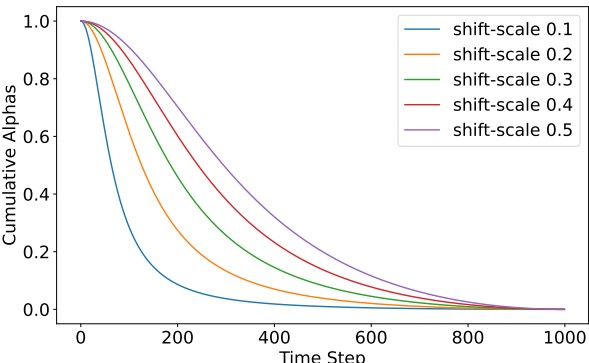

Figure 5: Visualization of our noise scheduler with various shift-scale factors. The $x$-axis represents the diffusion time steps, while the $y$-axis shows the cumulative alphas (Ho et al., 2020) of the noise scheduler.

SMOS results improve, the gap with GT in CMOS decreases, and the gap with the baseline gradually widens.

### A.6 NOISE SCHEDULE AND CLASSIFIER-FREE GUIDANCE

**Noise Scheduler** We use the cosine noise schedule (Nichol & Dhariwal, 2021) with scale-shift (Lovelace et al., 2024; Hoogeboom et al., 2023). Since the impact of noise schedules can vary depending on the resolution of the input, it may be necessary to shift the schedule to appropriately add noise across time steps. We test five different values of scale-shift and visualize them in Figure 5.

**Classfier-Free Guidance** To leverage classifier-free guidance (Ho & Salimans, 2021) for its efficacy in diffusion-based generative modeling, we train both a conditional and an unconditional diffusion model simultaneously. This is achieved by omitting the text input with a probability of $p = 0.1$. In the cross-attention layers of DiTTo, we concatenate a learnable null embedding with the text features along the sequence dimension, following the approach of (Lovelace et al., 2024). We eliminate the conditioning information by masking out the text embeddings during the cross-attention computation and setting the mean-pooled text embedding to zero.

**Hyperparameter Search for Noise Schedule and Sampling** We conduct a hyperparameter search for three components regarding noise schedule and sampling: (a) the scale-shift of the noise scheduler by training five different models with scales ranging from 0.1 to 0.5 (Figure 6a) as visualized in Figure 5, (b) the classifier-free guidance (CFG) scale during inference (Figure 6b), and (c) the number of diffusion steps during inference (Figure 6c). In these figures, The blue *dashed* line with circle marker represents the WER, and the green *dotted* line with square marker represents the SIM-r. The selected values in each experiment are marked with vertical black dash-dot

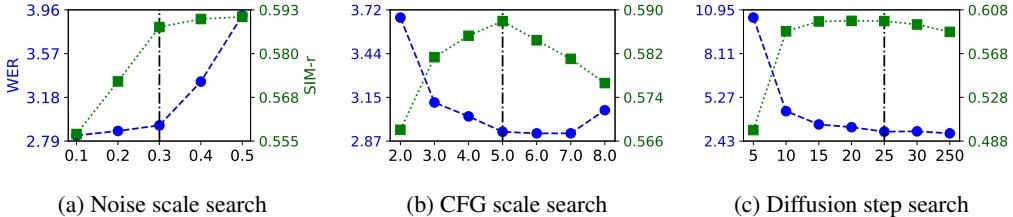

(a) Noise scale search        (b) CFG scale search        (c) Diffusion step search

Figure 6: A configuration search results for (a) scale-shift of the noise scheduler, (b) classifier-free guidance (CFG) scale, and (c) diffusion step search. We can conclude to set scale-shift to 0.3 from (a) and CFG scale to 5.0 from (b), and diffusion steps to 25 from (c).

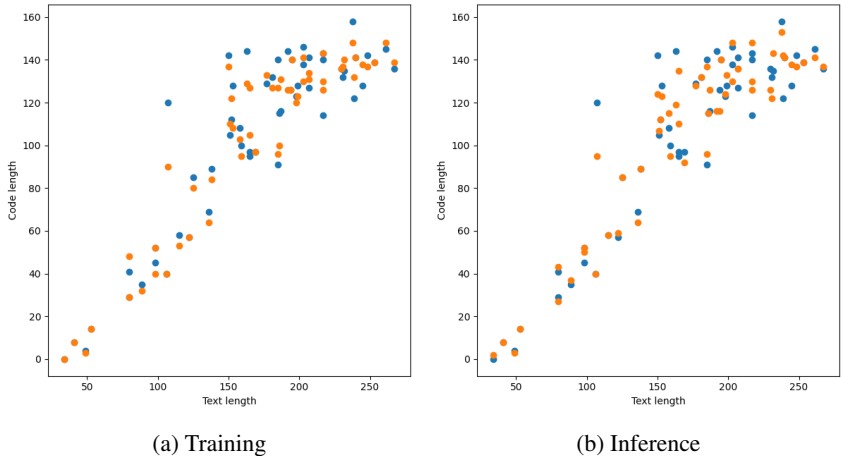

(a) Training                (b) Inference

Figure 7: Visualization of outputs with the text input and approximately 3 seconds of speech prompt from the speech length predictor during (a) training and (b) inference. In both plots, the $x$-axis represents the length of text tokens, and the $y$-axis represents the length of speech tokens. The blue dots indicate the ground truth lengths of speech tokens for each text input, while the orange dots represent the lengths predicted by the speech length predictor.

lines. Notably, 25 diffusion steps are sufficient to produce high-fidelity outputs, and we observe that performance gaps among different steps diminish with further training.

## A.7 TRAINING AND INFERENCE OF SPEECH LENGTH PREDICTOR

**Training** We train two speech length predictors for DiTTo-en and DiTTo-multi. Each was trained using the same text encoder as the respective DiTTo models. Essentially, we followed the same configuration as mentioned above, but without gradient accumulation and with a maximum token size of 10,240 for 600K steps. The number of trainable parameters for the speech length predictors is 33.18M for DiTTo-en and 33.58M for DiTTo-multi. Figure 7 presents a visualization of length sampling during training.

**Inference** After training the speech length predictor, inference can be performed either by calculating the expected value over all possible length values using its softmax logits with a maximum length of $N$ or by sampling from a multinomial distribution based on the same softmax logits. We set $N = 2048$. In both cases, we apply top-$k$ sampling (Fan et al., 2018) with $K = 20$. For objective and subjective evaluation, we use the expected length of the test samples. When no speech prompt is provided, the speech length predictor estimates the entire audio length starting from the '<BOS>' token, and the sampling process proceeds in the same manner as described earlier.

The speech length predictor is trained with causal masking to determine the total length based solely on a given speech prompt. During inference, however, it operates in a single forward pass, making it

Table 11: Performances for the English-only *cross-sentence* task. The value for DiTTo-en-S and DiTTo-en-XL is taken from Table 2. The boldface indicates the best result, the underline denotes the second best.

| Model | WER ↓ | CER ↓ | SIM-o ↑ | SIM-r ↑ |
|---|---|---|---|---|
| StyleTTS 2 (Li et al., 2024) | **2.53** | **0.77** | 0.3746 | - |
| XTTS-v2 (Casanova et al., 2024) | 5.05 | 1.74 | 0.4928 | - |
| DiTTo-en-S | 3.07 | 1.08 | 0.4984 | 0.5373 |
| DiTTo-en-XL | 2.56 | 0.89 | **0.6270** | **0.6554** |

Table 12: Human evaluations on comparison of demo samples from various SOTA models with DiTTo-en (XL). SMOS scores include a 95% confidence interval. The boldface indicates the best result.

| Model | SMOS | CMOS |
|---|---|---|
| Voicebox (Le et al., 2023) | $2.87_{\pm 0.45}$ | **0.14** |
| DiTTo-en | $\mathbf{3.57}_{\pm \mathbf{0.39}}$ | 0.0 |
| VALL-E (Wang et al., 2023) | $3.50_{\pm 0.46}$ | -0.94 |
| DiTTo-en | $\mathbf{3.90}_{\pm \mathbf{0.38}}$ | **0.0** |
| MegaTTS (Jiang et al., 2023) | $3.76_{\pm 0.44}$ | -0.31 |
| DiTTo-en | $\mathbf{3.82}_{\pm \mathbf{0.32}}$ | **0.0** |
| E3-TTS (Gao et al., 2023) | $4.25_{\pm 0.19}$ | -0.30 |
| DiTTo-en | $\mathbf{4.28}_{\pm \mathbf{0.32}}$ | **0.0** |
| NaturalSpeech 2 (Shen et al., 2024) | $3.99_{\pm 0.37}$ | -0.29 |
| DiTTo-en | $\mathbf{4.01}_{\pm \mathbf{0.35}}$ | **0.0** |
| NaturalSpeech 3 (Ju et al., 2024) | $\mathbf{4.42}_{\pm \mathbf{0.28}}$ | -0.27 |
| DiTTo-en | $3.92_{\pm 0.27}$ | **0.0** |

non-autoregressive. This design enables the predictor to learn the remaining lengths for all positions simultaneously, improving training efficiency and eliminating constraints on prompt length.

## A.8 COMPARISON WITH OPEN-SOURCE MODELS

We conducted an English-only *cross-sentence* task on StyleTTS 2 (Li et al., 2024) and XTTS-v2 (Casanova et al., 2024), with the results shown in Table 11. Our model significantly outperformed StyleTTS 2 in SIM-o, despite StyleTTS 2 having slightly higher WER and CER due to its use of phonemes and phoneme-level duration. XTTS-v2 also did not perform as well as DiTTo-en-S. These results underscore the favorable performance of our models compared to the others.

## A.9 HUMAN EVALUATIONS ON COMPARISON OF DEMO SAMPLES

Table 12 shows the results of human evaluation on demo cases. DiTTo outperforms all others except for NaturalSpeech 3 in SMOS and Voicebox in CMOS. We hypothesize that the prompts used in Voicebox tends to be unintelligible and noisy, and as shown in SMOS, the model samples did not adhere to them well. In contrast, DiTTo follows the prompts better, although the naturalness of the samples themselves is somewhat lacking. Therefore, in CMOS, when comparing two samples without prompts, it is likely that DiTTo will be judged as less natural. Conversely, in NaturalSpeech 3, as shown in SMOS, the model follows the intonation of the prompts too closely, sometimes resulting in unnaturalness. Thus, when comparing samples without prompts, DiTTo may sound more natural.

Table 13: Human evaluations on comparison of demo samples from concurrent works with DiTTo-en (XL). SMOS scores include a 95% confidence interval. The boldface indicates the best result.

| Model | SMOS | CMOS |
|---|---|---|
| SimpleSpeech (Yang et al., 2024a) | $3.50_{\pm 0.44}$ | -0.14 |
| DiTTo-en | $\mathbf{4.56_{\pm 0.19}}$ | **0.0** |
| E2 TTS (Eskimez et al., 2024) | $\mathbf{4.51_{\pm 0.16}}$ | -0.01 |
| DiTTo-en | $4.21_{\pm 0.29}$ | **0.0** |

Table 14: Human evaluations on comparison of RAVDESS Benchmark demo samples from various SOTA models with DiTTo-en (XL). SMOS scores include a 95% confidence interval. The boldface indicates the best result. For Voicebox (R) and VALL-E (R), the samples come from the reproduced models provided by NaturalSpeech 3 (Ju et al., 2024).

| Model | SMOS | CMOS |
|---|---|---|
| NaturalSpeech 2 (Shen et al., 2024) | $2.25_{\pm 0.70}$ | -1.07 |
| DiTTo-en | $\mathbf{3.75_{\pm 0.31}}$ | **0.0** |
| Voicebox (R) | $2.40_{\pm 0.59}$ | -1.26 |
| DiTTo-en | $\mathbf{4.10_{\pm 0.28}}$ | **0.0** |
| VALL-E (R) | $2.62_{\pm 0.58}$ | -1.39 |
| DiTTo-en | $\mathbf{3.79_{\pm 0.29}}$ | **0.0** |
| Mega-TTS 2 (Jiang et al., 2024) | $3.48_{\pm 0.40}$ | -0.80 |
| DiTTo-en | $\mathbf{3.83_{\pm 0.28}}$ | **0.0** |
| StyleTTS 2 (Li et al., 2024) | $2.79_{\pm 0.54}$ | -0.80 |
| DiTTo-en | $\mathbf{3.97_{\pm 0.38}}$ | **0.0** |
| HierSpeech++ (Lee et al., 2023a) | $2.33_{\pm 0.61}$ | -1.58 |
| DiTTo-en | $\mathbf{3.86_{\pm 0.33}}$ | **0.0** |
| NaturalSpeech 3 (Ju et al., 2024) | $3.86_{\pm 0.34}$ | -0.40 |
| DiTTo-en | $\mathbf{3.89_{\pm 0.25}}$ | **0.0** |

### A.10 HUMAN EVALUATIONS ON COMPARISON WITH CONCURRENT WORKS

We conduct an English-only *cross-sentence* task using SimpleSpeech (Yang et al., 2024a) and E2 TTS (Eskimez et al., 2024), with the results shown in Table 13, where all samples are at 16,000 Hz. Our model demonstrates favorable performance in all cases, except for the SMOS score of E2 TTS.

### A.11 HUMAN EVALUATIONS ON COMPARISON OF OUT-OF-DOMAIN RAVDESS DEMO SAMPLES

For the RAVDESS evaluation, we use NaturalSpeech 3 (Ju et al., 2024) demo samples at 16,000 Hz and measure CMOS and SMOS, as shown in Table 12, with our model's output downsampled to 16,000 Hz. Table 14 and Table 15 present results from the "Comparison Results on RAVDESS Benchmark" and "Zero-Shot TTS Samples (Emotion)" sections of the NaturalSpeech

Table 15: Human evaluations on comparison of RAVDESS Zero-Shot (Emotion) demo samples from various SOTA models with DiTTo-en (XL). SMOS scores include a 95% confidence interval.

| Model | SMOS | CMOS |
|---|---|---|
| NaturalSpeech 3 (Ju et al., 2024) | $\mathbf{4.17_{\pm 0.26}}$ | -0.62 |
| DiTTo-en | $4.14_{\pm 0.24}$ | **0.0** |

Table 16: Performances of DiTTo-multi for the multilingual *continuation* task.

| Model | WER ↓ | CER ↓ | SIM-o ↑ | SIM-r ↑ |
|---|---|---|---|---|
| English / MLS English | 6.91 | 4.15 | 0.4759 | 0.4986 |
| English (HuBERT) / MLS English | 5.73 | 1.84 | - | - |
| German / MLS German | 6.60 | 2.47 | 0.4917 | 0.5239 |
| Dutch / MLS Dutch | 8.89 | 3.11 | 0.5828 | 0.5971 |
| French / MLS French | 8.03 | 3.09 | 0.5711 | 0.5905 |
| Spanish / MLS Spanish | 2.22 | 0.89 | 0.5483 | 0.5776 |
| Italian / MLS Italian | 13.69 | 2.35 | 0.5637 | 0.5902 |
| Portuguese / MLS Portuguese | 6.07 | 2.11 | 0.5346 | 0.5289 |
| Polish / MLS Polish | 10.88 | 3.17 | 0.5090 | 0.5441 |
| Korean / AIHub 14 | 19.44 | 1.90 | 0.5838 | 0.6095 |
| Korean / AIHub 15 | 13.80 | 2.20 | 0.5260 | 0.5454 |
| Korean / Ksponspeech | 27.22 | 18.30 | 0.5205 | 0.5466 |

Table 17: Performances of CLaM-multi for the multilingual *continuation* task.

| Model | WER ↓ | CER ↓ | SIM-o ↑ |
|---|---|---|---|
| English / MLS English | 8.71 | 5.19 | 0.4000 |
| English (HuBERT) / MLS English | 7.71 | 3.19 | 0.4000 |
| German / MLS German | 9.63 | 4.11 | 0.4219 |
| Dutch / MLS Dutch | 12.25 | 4.97 | 0.5983 |
| French / MLS French | 10.29 | 4.08 | 0.5671 |
| Spanish / MLS Spanish | 4.02 | 1.91 | 0.5292 |
| Italian / MLS Italian | 19.70 | 5.19 | 0.5459 |
| Portuguese / MLS Portuguese | 9.66 | 3.72 | 0.5658 |
| Polish / MLS Polish | 14.70 | 5.34 | 0.5519 |
| Korean / AIHub 14 | 20.21 | 1.80 | 0.5423 |
| Korean / AIHub 15 | 13.08 | 2.35 | 0.5280 |
| Korean / Ksponspeech | 30.24 | 20.02 | 0.4488 |

3 demo page[19], respectively. In the RAVDESS benchmark, our model outperforms all baselines and demonstrates MOS scores comparable to NaturalSpeech 3, along with superior CMOS performance in additional zero-shot emotion settings. This trend is consistent with the results shown in Table 12, and we interpret these findings similarly to our previous experiments. These results indicate the strong performance of DiTTo, even with out-of-domain audio prompts.

### A.12 MULTILIGUAL CONTINUATION TASK

In this section, we provide the experimental results of multilingual continual tasks for both DiTTo-multi in Table 16 and CLaM-multi in Table 17.

### A.13 SCALING DATA

For the two experiments, we construct each data subset as follows: 0.5K from LibriTTS (Zen et al., 2019) only, 5.5K from LibriTTS and MLS (Pratap et al., 2020) English subset 5K, 10.5K from LibriTTS and MLS English subset 10K, and 50.5K from LibriTTS and MLS English subset 50k, in hours. The evaluation is conducted on the English-only *cross-sentence* task. To ensure fast convergence of Mel-VAE, we replace the commitment loss with the original RVQ (Lee et al., 2022), as used in other audio codecs (Défossez et al., 2023; Kumar et al., 2024), where the commitment loss is calculated at each depth. We also measure PESQ (Rix et al., 2001) and ViSQOL (Chinen et al., 2020) for different Mel-VAEs. Please refer to Appendix A.3 for metric details.

---

[19]https://speechresearch.github.io/naturalspeech3/

Table 18: Performances of the English-only *cross-sentence* task along with data scales, given the same Mel-VAE (from *libritts+mls-en-50k* of Table 19) latents as the target. The boldface highlights the best result, with scales increasing from top to bottom.

| Model | WER ↓ | SIM-r ↑ | Codec PESQ ↑ | Codec ViSQOL ↑ |
|---|---|---|---|---|
| DiTTo-*libritts* | 3.30 | 0.5782 | | |
| DiTTo-*libritts+mls-en-5k* | 3.14 | 0.5664 | 2.68 | 4.56 |
| DiTTo-*libritts+mls-en-10k* | 2.97 | 0.5704 | | |
| DiTTo-*libritts+mls-en-50k* | **2.89** | **0.5783** | | |

Table 19: Performances of the English-only *cross-sentence* task along with data scales, when the paired Mel-VAE and DiTTo are trained on the same data subset. The boldface highlights the best result, with scales increasing from top to bottom.

| Model | WER ↓ | SIM-r ↑ | Codec PESQ ↑ | Codec ViSQOL ↑ |
|---|---|---|---|---|
| *libritts* | 3.49 | 0.5561 | **2.75** | **4.60** |
| *libritts+mls-en-5k* | 2.95 | 0.5552 | 2.67 | 4.57 |
| *libritts+mls-en-10k* | 2.91 | 0.5706 | 2.65 | 4.55 |
| *libritts+mls-en-50k* | **2.89** | **0.5783** | 2.68 | 4.56 |

The first experimental results are presented in Table 18, showing that: (1) Even with the 0.5K dataset, the model exhibits acceptable speech accuracy (WER=3.30) and speaker similarity to the prompt (SIM-r=0.5782). At 5.5K, there is an improvement in speech accuracy (WER=3.14). (2) Beyond 5.5K, further increases in dataset size result in further improvements, with speaker similarity showing similar scores or incremental gains. (3) We observe that models trained on smaller datasets like 0.5K show lower speech accuracy compared to phoneme-level duration-based models such as P-Flow (Kim et al., 2024b). This is one of the limitations of our model.

Table 19 presents the second experimental results. Compared to the first experiment, performance drops with a smaller dataset, especially at 0.5K, even though Codec PESQ and ViSQOL scores improve compared to larger datasets. As the dataset size increases, the benefits of data scaling become more apparent, particularly in terms of WER. This pattern aligns with the data scaling results in NaturalSpeech 3 (Ju et al., 2024).

We further evaluate our model under more restricted conditions, using a dataset as limited as the 60 hours used by Voicebox in Section B.2 (Le et al., 2023). Specifically, we train a DiTTo variant on approximately 50 hours of data by randomly sampling 10% of the LibriTTS dataset, following the same experimental settings as the first experiment in Table 18. However, due to the limited data, we encounter overfitting and apply early stopping at 40K steps (instead of the planned 200K steps). At this point, the model achieves a WER of 9.25 and a SIM-r of 0.3336. Although these results do not allow for a direct comparison with Voicebox's corresponding setting due to overfitting, we believe that incorporating phoneme-level durations during training, as done in Voicebox, ensures greater robustness when working with small datasets.

## A.14 SPEECH LENGTH MODELING COMPARISON

For evaluation, the test set is composed of five randomly selected samples per speaker from 42 speakers in the test set of the MLS English subset. The results in Table 5 show how variable length modeling in diffusion models (referred to as *SLP-CE* and *SLP-Regression*) for TTS enhances performance compared to fixed length modelings (referred to as *fixed-length-full* and *fixed-length*). *fixed-length* includes only a portion of the padding in the loss, similar to Simple-TTS (Lovelace et al., 2024), while *fixed-length-full* includes the entire padding. We observed that as the proportion of padding included increases, performance decreases. The key point is that excluding padding in variable length modeling is essential for improving the performance of diffusion models, rather than including padding in the target distribution as in existing fixed length modeling approaches.

Table 20: Performances of English-only cross-sentence task for Mel-VAE++. The boldface indicates the best result.

| DiTTo Text Encoder | Mel-VAE++ | WER ↓ | CER ↓ | SIM-o ↑ | SIM-r ↑ |
|---|---|---|---|---|---|
| SpeechT5 | - | 3.07 | 1.15 | 0.5423 | 0.5858 |
| | HuBERT | 5.59 | 3.18 | 0.5239 | 0.5742 |
| | SpeechT5 | 3.09 | 1.12 | 0.5335 | 0.5940 |
| | ByT5-base | 2.99 | 1.06 | 0.5364 | **0.5982** |
| | ByT5-base + HuBERT | **2.93** | **1.04** | **0.5495** | 0.5958 |
| ByT5-base | - | 6.22 | 3.82 | 0.5482 | 0.5945 |
| | HuBERT | 5.88 | 3.64 | 0.5458 | 0.5938 |
| | SpeechT5 | 4.10 | 1.95 | 0.5465 | 0.6055 |
| | ByT5-base | **3.11** | **1.17** | 0.5323 | 0.5965 |
| | ByT5-base + HuBERT | 3.69 | 1.72 | **0.5631** | **0.6101** |
| ByT5-large | - | 3.92 | 1.84 | 0.5292 | 0.5756 |
| | HuBERT | **3.04** | **1.11** | 0.5429 | 0.5878 |
| | SpeechT5 | 3.27 | 1.31 | 0.5412 | **0.6029** |
| | ByT5-base | 3.17 | 1.18 | 0.5368 | 0.5999 |
| | ByT5-base + HuBERT | 3.80 | 1.75 | **0.5489** | 0.5975 |

## A.15 MEL-VAE++

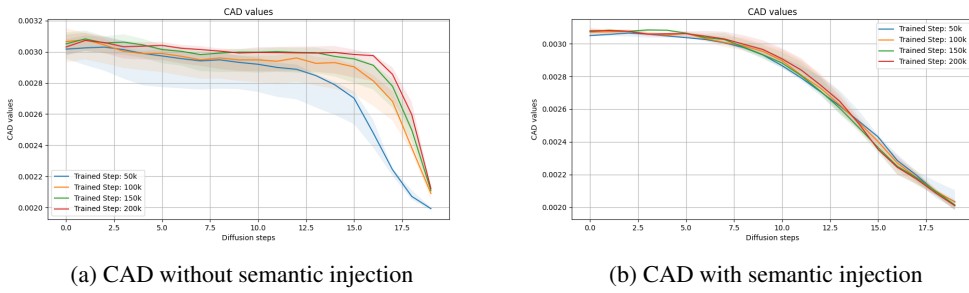

(a) CAD without semantic injection

(b) CAD with semantic injection

Figure 8: Comparison of CAD between training with and without semantic injection

When Mel-VAE is finetuned with the guidance of a pre-trained language model, consistent improvement in performance is observed, closing the gap with the SpeechT5 encoder, which is trained on both unsupervised speech and language data. To improve our understanding of semantic content injection, we finetune Mel-VAE with various pre-trained text and speech encoder models. As shown in Table 20, the experiments use various combinations of DiTTo text encoders and Mel-VAE text encoders. When SpeechT5 is utilized as the text encoder for DiTTo, semantic content injection enhances the performance metrics; however, its gains, while positive, are not as substantial as those observed in the control, which is trained the same number of steps without auxiliary signal, denoted as '-'. When ByT5-base is used as the DiTTo text encoder, using ByT5-base as Mel-VAE semantic content injection model significantly boosts metrics. We additionally observe that HuBERT feature matching, as done in (Zhang et al., 2024), helps refine its speech quality, as can be seen in the improved Speaker Similarity (SIM) metrics. When ByT5-base and HuBERT are both used, Mel-VAE benefits from both models.

We also investigate how DiTTo interacts with the text encoder latents as training progresses, in settings with and without semantic content injection. Since the TTS process typically ensures a monotonic alignment between text and speech, the diagonality of cross-attention between the text encoder and DiT block indicates how effectively text latents are used during denoising process. The intensity of the cross-attention map reflects how conducive the information provided by the text

encoder is to producing DiTTo's output. We measure the monotonicity of cross-attention maps using Cumulative Attention Diagonality (CAD) (Shim et al., 2022).

The median CAD values are drawn in solid lines as in Figure 8. The model performs inference on 5 samples. The Interquartile Range (IQR) is represented by the shaded regions surrounding the solid lines in the graph. We set the sampling steps as 25, noise schedule scale shift as 0.3, and classifier-free guidance scale as 5.0. Given that the parameters of text encoders are not tuned during DiTTo training, drastic changes in CAD values across training steps without semantic content injection indicate significant changes in DiTTo parameters are needed to align with text encoders. In contrast, with semantic content injection, changes in CAD values as training progresses show minimal fluctuation. This suggests that semantic content injection enables effective interaction between text encoders and DiT blocks. Based on CAD values across diffusion steps, semantic content injection allows the interaction between the text encoder and DiTTo to predominantly occur during the initial stages of the reverse process. This suggests that semantic content injection boosts the provision of rich semantic information by text encoders, allowing DiT blocks to focus on improving acoustic quality in later steps. Furthermore, the significant deviations across samples, as evidenced by the enlarged shaded regions when semantic injection is not employed, suggest that the interaction between the text encoder and DiT blocks is less effective without semantic content injection.

## A.16 TRAINING SETUP FOR KEY ASPECTS EXPERIMENTS

All models are trained for 200K steps using the MLS English subset in the base (B) configuration, with SpeechT5 as the text encoder and the original Mel-VAE as the audio latent codec. We adopt the same span masking strategy, utilize L1 loss, and limit the training data to 20 seconds, as described in (Lovelace et al., 2024). Other hyperparameters remain consistent with those used in DiTTo's main training. The total training time is set to one day.

## A.17 MEANING OF LATENT QUANTIZATION IN LDM-BASED TTS

We use the latent representation of Mel-VAE before quantization as the target latent for DiTTo. According to CLaM-TTS (Kim et al., 2024a), probabilistic RVQ is the key factor that enables Mel-VAE to achieve significant length compression in the time domain compared to existing codecs. This allows Mel-VAE to generate an effective latent with a short sequence length while maintaining high-quality speech reconstruction. Quantization helps mitigate issues related to the KL loss weight coefficient during VAE training (Gray, 1984). In the audio domain, RVQ is preferred over VQ due to the higher sampling rates, as demonstrated in models like EnCodec (Défossez et al., 2023) and DAC (Kumar et al., 2024). However, as we confirm through the experiments in Section 6, quantization is crucial for generating a good latent representation within the autoencoder, but for an LDM like DiTTo, which uses these latents as a target, quantization itself is irrelevant.

## A.18 IMPACT OF TRAINING STEPS ON PERFORMANCE

Table 21: Performance at different training steps of DiTTo-*mls* from Section 5.1. The boldface indicates the best result.

| Training Steps | WER ↓ | SIM-r ↑ |
|---|---|---|
| *50K* | 7.27 | 0.5225 |
| *100K* | 3.51 | 0.5630 |
| *150K* | 3.17 | 0.5770 |
| *200K* (from Table 4) | **2.93** | **0.5877** |

We conduct two evaluations to investigate the impact of training steps on *cross-sentence* task performance. Table 21 presents the results of evaluating the performance of DiTTo-*mls* from Section 5.1 across different training steps. Specifically, we assess the model's performance at 50K, 100K, 150K, and 200K training steps to analyze training dynamics. Our findings reveal that the DiTTo-*mls* model achieves reasonable WER and SIM performance as early as 100K steps.

Table 22: Trade-off between WER and SIM performance for DiTTo-*dac-24k* and DiTTo-*dac-44k* models with varying scale-shift values. Configurations are labeled as DiTTo-*dac-\*-scale-shift-\**, where the first placeholder indicates the sampling rate (24K or 44K) and the second denotes the scale-shift value in the noise scheduler. Codec PESQ and ViSQOL scores are included for codec quality comparison.

| Model | WER ↓ | SIM-o ↑ | SIM-r ↑ | Codec PESQ ↑ | Codec ViSQOL ↑ |
|---|---|---|---|---|---|
| DiTTo-*dac-24k-scale-shift-0.1* | 4.78 | 0.5042 | 0.5098 | | |
| DiTTo-*dac-24k-scale-shift-0.2* | 4.70 | 0.5302 | 0.5362 | 4.37 | 4.91 |
| DiTTo-*dac-24k-scale-shift-0.3* | 7.21 | 0.5478 | 0.5545 | | |
| DiTTo-*dac-44k-scale-shift-0.1* | 7.70 | 0.5197 | 0.5366 | | |
| DiTTo-*dac-44k-scale-shift-0.2* | 8.71 | 0.5237 | 0.5428 | 3.74 | 4.85 |
| DiTTo-*dac-44k-scale-shift-0.3* | 14.58 | 0.5391 | 0.5597 | | |
| DiTTo-*mls* (from Table 4) | 2.93 | 0.5467 | 0.5877 | 2.95 | 4.66 |

Table 23: Performances of Mel-VAE with different time-domain compression ratios. The training and evaluation settings are consistent with those outlined in Section 5.2. $2\times$, $4\times$, and $8\times$ represent the compression ratios in the time-domain, with $8\times$ corresponding to the original Mel-VAE configuration used in our paper.

| Model | WER ↓ | SIM-o ↑ | SIM-r ↑ | Codec PESQ ↑ | Codec ViSQOL ↑ |
|---|---|---|---|---|---|
| DiTTo-*Mel-VAE*-$2\times$ | 5.35 | 0.5525 | 0.5931 | | |
| DiTTo-*Mel-VAE*-$4\times$ | 3.44 | 0.5481 | 0.5867 | - | - |
| DiTTo-*Mel-VAE*-$8\times$ | 3.14 | 0.5337 | 0.5774 | | |
| Mel-VAE-$2\times$ | | | | 2.70 | 4.54 |
| Mel-VAE-$4\times$ | - | - | - | 2.72 | 4.57 |
| Mel-VAE-$8\times$ | | | | 2.74 | 4.58 |

## A.19 TRADE-OFF BETWEEN WER AND SIM PERFORMANCE IN DiTTo-DAC

DiTTo-*dac-24k* shows significantly worse WER performance but only slightly higher SIM-o compared to DiTTo-*mls*, as presented in Table 7 in Section 6. However, the trade-off trend observed in Figure 6a suggests that optimizing WER inevitably results in a decline in SIM. This raises the question of whether DiTTo-*dac-24k* maintains better SIM-o performance than DiTTo-*mls*, even as WER improves. To investigate this, we conduct experiments by ablating scale-shift values of 0.1, 0.2, and 0.3 to optimize WER while monitoring changes in SIM (including evaluations for DiTTo-*dac-44k*). The results, summarized in Table 22, confirm a clear trade-off between WER and SIM: reducing the scale-shift value improves WER but lowers SIM. For instance, decreasing the scale-shift value for DiTTo-*dac-24k* from 0.3 to 0.2 improves WER from 7.21 to 4.70, though still suboptimal, while SIM-o drops from 0.5478 to 0.5302. At a scale-shift value of 0.1, most diffusion timesteps become excessively noisy, negatively impacting both WER and SIM. Therefore, even after optimization, DiTTo-*dac-24k* does not surpass DiTTo-*mls* in either WER or SIM performance.

## A.20 PERFORMANCE OF MEL-VAE VARIANTS WITH DIFFERENT COMPRESSION RATIOS

Table 23 presents the results of Mel-VAE variants with different compression ratios (2×, 4×, and 8×, as in the original Mel-VAE). We conclude that performance improves as the compression ratio increases (i.e., as the latent length decreases). We apply the same commitment loss modification described in Appendix A.13 for faster convergence.

Table 24: Comparison of DiTTo variants trained on different target representations, including mel-spectrograms (DiTTo-*mel*) and latent representations (DiTTo-*encodec* and DiTTo-*mls*). The sequence length in the time domain differs across models, with DiTTo-*mls* having the greatest compression, followed by DiTTo-*encodec*, and then DiTTo-*mel*. The boldface indicates the best result for each metric.

| Model | WER ↓ | SIM-o ↑ | SIM-r ↑ | SMOS | CMOS |
|---|---|---|---|---|---|
| DiTTo-*mls* (from Table 4) | **2.93** | 0.5467 | **0.5877** | **3.67**±**0.14** | **0.00** |
| DiTTo-*encodec* (from Table 7) | 4.19 | 0.5105 | 0.5460 | 2.86±0.15 | -1.43 |
| DiTTo-*mel* | 4.65 | **0.5637** | 0.5792 | 3.58±0.14 | -0.44 |

## A.21 TARGETING MEL-SPECTROGRAMS WITHOUT LATENT MODELINGS

We choose to use a latent representation from Mel-VAE instead of directly modeling mel-spectrograms because compact representations in the time domain improve diffusion model performance, particularly in intelligibility, as demonstrated by the WER results in Appendix A.20. To support this, we conduct additional experiments comparing DiTTo-*mel* (trained directly on mel-spectrograms) with DiTTo-*encodec* and DiTTo-*mls* (trained on latent representations). The results in Table 24 indicate that greater compression (DiTTo-*mls* > DiTTo-*encodec* > DiTTo-*mel*) improves WER performance by shortening the sequence length in the time domain, consistent with the findings in Appendix A.20.

DiTTo-*mel* achieves the highest SIM-o score by directly predicting ground truth mel-spectrograms and avoiding the information loss typically introduced by autoencoder latent decoding. However, the overall performance of DiTTo-*mls* is comparable when considering both SIM-o and SIM-r together, while showing noticeably better intelligibility than DiTTo-*mel*. Both SMOS and CMOS subjective evaluations confirm that DiTTo-*mls* outperforms the other models, validating the objective results. The notably poor performance of DiTTo-*encodec* appears to stem from the relatively lower quality of the EnCodec (Défossez et al., 2023). As shown in Table 1 and Section 5.1, DiTTo-*mls* also offers fast inference speeds and remains faster than Voicebox (Le et al., 2023), which directly models mel-spectrograms, even with the same number of diffusion steps. This highlights the effectiveness of Mel-VAE as a target latent representation, balancing intelligibility, speech similarity, and efficiency.

