# OpenReview forum: "DiTTo-TTS: Diffusion Transformers for Scalable Text-to-Speech without Domain-Specific Factors"
_ICLR.cc/2025/Conference — ICLR 2025 Poster_

### Official Review · Reviewer_Tbzi · 2024-10-31

**Soundness:** 3
**Presentation:** 3
**Contribution:** 3
**Rating:** 6
**Confidence:** 4

**Summary:**

This study presents a model for personalized speech synthesis based on an LDM structure. Personalization is achieved by modeling z obtained through Mel-VAE using an infilling approach. Instead of explicitly calculating text-speech alignment and providing text information accordingly, as is common in previous research, the model allows the text and speech representations to learn alignment autonomously via cross-attention. Additionally, a variable-length predictor is introduced to predict the total speech length during inference. To validate the proposed structure and methodology, the authors conducted comparisons with numerous strong baselines across various languages and benchmarks, as well as extensive ablation studies on proposed elements, data quantity, hyperparameters, and other factors.

**Strengths:**

The most notable strength is that the authors conducted comparisons with various baselines and performed extensive ablation studies and analyses on numerous elements. In addition, the paper is well-written, with a high level of detail in the experiments, making it easy to understand the study.

**Weaknesses:**

Personally, I feel that the novelty is somewhat limited, as LDM-based structures in zero-shot TTS are not entirely new (e.g., NaturalSpeech 2, 3, as you already mentioned in your paper.), and the approach of allowing the model to learn speech-text alignment with cross-attention has been seen in previous works, such as Simple-TTS. Nonetheless, the systematic analysis of each component and the guidance provided to readers on existing design choices and their limitations are indeed valuable contributions.

**Questions:**

1. Are you plan to release the code and checkpoints of DiTTo-TTS?

2. Regarding the content in Appendix A.13, I am interested in whether the model performs well in smaller data scenarios, such as a single-speaker setup with less data than LibriTTS. The scale-up trend in performance relative to data quantity seems less pronounced compared to other zero-shot TTS models (e.g., Voicebox Section B.2). I would like to understand the authors' perspective on why DiTTo-TTS remains robust even with limited data.

---

> ### Author Response · Authors · 2024-11-19
> **Authors' Response to Reviewer Tbzi**
>
> We sincerely appreciate your encouraging review and constructive feedback, and we hope our response thoroughly resolves your concerns.
>
> `[W1] Concern on novelty with acknowledgment of valuable analysis`
>
> > Personally, I feel that the novelty is somewhat limited, as LDM-based structures in zero-shot TTS are not entirely new (e.g., NaturalSpeech 2, 3, as you already mentioned in your paper.), and the approach of allowing the model to learn speech-text alignment with cross-attention has been seen in previous works, such as Simple-TTS. Nonetheless, the systematic analysis of each component and the guidance provided to readers on existing design choices and their limitations are indeed valuable contributions.
> >
>
> We are grateful for your kind words and acknowledgment of the significance of our contributions. Regarding the novelty, as you rightly pointed out, LDM-based TTS systems are not entirely new, and we would like to clarify that we are not claiming to be the first to explore that. We also give full credit to previous works like Simple-TTS for demonstrating the possibility of LDM-based TTS without domain-specific factors. However, as you have kindly recognized, our main contribution is an experimental analysis of key aspects that enable such simple modeling to achieve state-of-the-art performance, validated by our final model's strong results, as detailed in our resaerch questions R1 and R2 of Section 5.
>
> `[Q1] Plans for code and checkpoint release of DiTTo-TTS`
>
> > 1. Are you plan to release the code and checkpoints of DiTTo-TTS?
> >
>
> We thank reviewer for interest in our open-sourcing plans. As mentioned in Section 9 (REPRODUCIBILITY STATEMENTS) of our paper, we plan to release the inference code, pre-trained weights, and eventually the full training implementation in phases, provided any legal issues are resolved.
>
> `[Q2] Robustness of DiTTo-TTS in low-data scenarios`
>
> > 2. Regarding the content in Appendix A.13, I am interested in whether the model performs well in smaller data scenarios, such as a single-speaker setup with less data than LibriTTS. The scale-up trend in performance relative to data quantity seems less pronounced compared to other zero-shot TTS models (e.g., Voicebox Section B.2). I would like to understand the authors' perspective on why DiTTo-TTS remains robust even with limited data.
> >
>
> We are grateful for your thoughtful feedback and for highlighting this important point. The performance improvement trend from 0.5K hours of data for *DiTTo-libritts* to 5.5K hours for *DiTTo-libritts+mls-en-5k* aligns with the trend observed in Voicebox from 0.6 to 6K hours (as discussed in Section B.2 of their paper), indicating similar patterns. To address your question, we need to evaluate our model under more restricted conditions, such as the 60 hours used by Voicebox.
>
> We trained a DiTTo-*libritts-0.05k* model on approximately 50 hours of data, using the same settings described in Appendix A.13, Table 18, by randomly sampling around 10% of the LibriTTS dataset. Due to the limited data, we encountered overfitting and applied early stopping at 40K steps (instead of the planned 200K steps). The results, summarized in the table below, show significantly lower performance. The performance of models other than DiTTo-*libritts-0.05k* was directly taken from Table 18 in the paper.
>
> | Model                          | WER ↓    | SIM-r ↑  |
> |--------------------------------|----------|----------|
> | DiTTo-*libritts-0.05k*         | 9.25     | 0.3336   |
> | DiTTo-*libritts*               | 3.30     | 0.5782   |
> | DiTTo-*libritts+mls-en-5k*     | 3.14     | 0.5664   |
> | DiTTo-*libritts+mls-en-10k*    | 2.97     | 0.5704   |
> | DiTTo-*libritts+mls-en-50k*    | 2.89     | 0.5783   |
>
> Although these results do not allow for a direct comparison with Voicebox’s corresponding setting due to overfitting, we believe that incorporating phoneme-level durations during training, as done in Voicebox, enhances robustness when working with small datasets. This is implicitly supported by our observation in the second paragraph (3) of Appendix A.13, where our WER was worse than that of P-Flow, which incorporates phoneme-level durations, when trained on 0.5K data.
>
> We include this explanation at the end of Appendix A.13 in the revised manuscript. We sincerely appreciate your suggestions, which have helped improve the quality of our work.

---

> > ### Comment · Reviewer_Tbzi · 2024-11-20
> > **Thank you for your kind response.**
> >
> > Thank you for your kind response.
> >
> > 1. Regarding the open-source plan, I sincerely hope that any legal issues are resolved smoothly so that this model can benefit researchers.
> >
> > 2. I also fully understand your explanation about the low-data scenario. To clarify my earlier question, which may have been phrased somewhat unclearly: while Voicebox demonstrates a consistent increase in SIM-r as data size grows (0.151 -> 0.417 -> 0.573 -> 0.645), DiTTo-TTS appears to achieve strong SIM-r values even with a small amount of data, reaching a saturation point (0.578 -> 0.566 -> 0.570 -> 0.578). I found this characteristic to be an advantage, which is why I brought it up.
> >
> > Once again, thank you for your thoughtful reply, and I will maintain my initial score.

---

> > > ### Author Response · Authors · 2024-11-21
> > > **Re: Thank you for your kind response.**
> > >
> > > We sincerely appreciate your kind response and thoughtful feedback.
> > >
> > > 1. We share your hope of contributing to the research community through open-sourcing and are working diligently to resolve any potential legal issues.
> > > 2. Thank you for rephrasing your question and highlighting an aspect that further underscores the advantages of our approach. We attribute this characteristic to the effectiveness of our latent representation compared to mel-spectrograms. Specifically, as shown in Table 7 and Section 6, Mel-VAE outperforms other latent representations and mel-spectrograms as a diffusion target, as discussed in [Q2] of the Authors’ Response to Reviewer sHU2. This effectiveness enables Mel-VAE’s decoder to generate high-quality outputs with sufficiently informative latent inputs, even under limited training steps.
> > >
> > > Once again, we sincerely thank you for your efforts in evaluating our work and helping us improve the paper. We would be happy to address any further questions or suggestions if you have.

---

### Official Review · Reviewer_jyAn · 2024-10-31

**Soundness:** 3
**Presentation:** 3
**Contribution:** 2
**Rating:** 5
**Confidence:** 4

**Summary:**

This paper introduces DiTTo-TTS, a scalable TTS model based on diffusion transformers that avoids reliance on domain-specific factors like phonemes or durations, achieving high-quality results and efficiency improvements over conventional TTS methods.

**Strengths:**

1. Develop a diffusion-based TTS model, DiTTo-TTS, that avoids reliance on domain-specific factors while maintaining competitive quality.
2. DiTTo-TTS is shown to achieve better performance compared to state-of-the-art baselines with significantly reduced inference time.

**Weaknesses:**

**Dependency on Domain-specific factors** DiTTo-TTS is claimed to avoid reliance on domain-specific factors like phonemes or durations; however it needs the pre-trained Text Encoder (ByT5, SpeechT5), which aligns the textual and acoustic information into a unified semantic space, and a pre-trained neural audio codec which aligns the speech latents with the linguistic latents of the pre-trained language model during the autoencoding process. Moreover, DiTTo-TTS also needs a Speech Length Predictor for managing duration. Given these dependencies, it’s unclear how the model avoids domain-specific factors.  Could you clarify your definition of "domain-specific factors" and explain how your approach differs from traditional TTS systems in terms of reliance on phonemes and durations, even with the use of pre-trained components?

**Limited Novelty of the Proposed Approach** Simple-TTS and NaturalSpeech 2 has already proposed to use a pre-trained text encoder and neural audio codec to align the speech with text through cross-attention and training only the weight of latent diffusion model. Could you provide a more detailed comparison between DiTTo-TTS and Simple-TTS/NaturalSpeech 2 and highlight the key innovations or improvements in DiTTo-TTS that differentiate it from these prior works?

**Questions:**

1. Can you provide confidence intervals or p-values for the comparisons in your result tables? Without these, it is impossible to tell if the result is significantly better than state-of-the-art baselines.

2. In Appendix A3, the instruction for comparative mean opinion score (CMOS) shows that you only ask participants to compare the quality of the synthesized audio and reference, but in the main text you mention that DiTTo-en significantly outperforms the baseline models in naturalness, quality, and intelligibility (as assessed by CMOS). This raises questions about how the evaluations of naturalness and intelligibility for the synthesized audio were conducted in the experiments.

**Details Of Ethics Concerns:**

Research involving human subjects

---

> ### Author Response · Authors · 2024-11-19
> **Authors' Response to Reviewer jyAn (1/2)**
>
> We are truly grateful for your thorough feedback and constructive input, which has helped refine the manuscript. Below, we provide detailed responses to each of your points.
>
> `[W1] Clarification of "domain-specific factors"`
>
> > **Dependency on Domain-specific factors** DiTTo-TTS is claimed to avoid reliance on domain-specific factors like phonemes or durations; however it needs the pre-trained Text Encoder (ByT5, SpeechT5), which aligns the textual and acoustic information into a unified semantic space, and a pre-trained neural audio codec which aligns the speech latents with the linguistic latents of the pre-trained language model during the autoencoding process. Moreover, ..., Could you clarify your definition of "domain-specific factors" and explain how your approach differs from traditional TTS systems in terms of reliance on phonemes and durations, even with the use of pre-trained components?
> >
>
> We thank the reviewer for insightful feedback and for allowing us the opportunity to clarify our work. First, we would like to note that ByT5 is a text-only pre-trained model, and that DiTTo-TTS can be trained using a text-only pre-trained encoder alongside any neural audio codec without requiring semantic injection. The absence of phonemes and phoneme-level durations enables the effective utilization of pre-trained text encoders, similar to text-to-image models that use pre-trained text encoders and image autoencoders. This illustrates that latent diffusion model-based TTS can be implemented using components analogous to frameworks in other domains, such as text-to-image.
>
> Moreover, improving speech-text alignment by enhancing pre-trained models for each modality is not a domain-specific methodology. We have found and empirically demonstrated that cross-attention training is enhanced when the alignment between the speech and text domains is strengthened, and this approach can be applied to any two domains. For example, methods like CLIP [1], which improve alignment between text and images, are not considered specific to the image domain. The Speech Length Predictor is one method for enabling variable-length modeling in diffusion processes and does not rely on phonemes or phoneme-level durations, unlike duration predictors in traditional TTS systems.
>
> We define "domain-specific factors" as phonemes and phoneme-level durations, which traditionally require specialized linguistic preprocessing and detailed alignment in TTS systems. Our approach minimizes reliance on these factors, setting it apart from traditional TTS systems and offering a more scalable and efficient solution.
>
> [1] Radford, A., Kim, J. W., Hallacy, C., Ramesh, A., Goh, G., Agarwal, S., ... & Sutskever, I. (2021, July). Learning transferable visual models from natural language supervision. In International conference on machine learning (pp. 8748-8763). PMLR.
>
> `[W2] Comparison of DiTTo-TTS with Simple-TTS and NaturalSpeech 2`
>
> > **Limited Novelty of the Proposed Approach** Simple-TTS and NaturalSpeech 2 has already proposed to , ..., Could you provide a more detailed comparison between DiTTo-TTS and Simple-TTS/NaturalSpeech 2 and highlight the key innovations or improvements in DiTTo-TTS that differentiate it from these prior works?
> >
>
> Thanks for your insightful comments and for the opportunity to clarify the distinctions between DiTTo-TTS and prior works like Simple-TTS and NaturalSpeech 2. Firstly, we would like to note that NaturalSpeech 2 does not utilize a pre-trained language model text encoder, as it relies on phoneme inputs. Additionally, NaturalSpeech 2 does not use cross-attention to align speech with text; instead, it uses phoneme-level durations to pre-align text representations with speech before applying them as conditioning inputs. Beyond this, NaturalSpeech 2 also requires pitch conditioning, which involves preprocessing steps to extract pitch information from the audio and depends on a third-party pitch extractor to obtain ground truth pitch values. In other words, traditional powerful TTS models require such TTS-specific factors, which increase model complexity and preprocessing costs, ultimately hindering scalability.
>
> Simple-TTS established that LDM-based TTS could be achieved without domain-specific factors, marking it as a pioneering effort in this emerging paradigm. However, it lagged significantly behind state-of-the-art performance, lacking insights into the causes of this gap. Our main contribution is an experimental analysis of key aspects that enable such simple modeling to achieve state-of-the-art performance, validated by our final model's strong results, as detailed in our resaerch questions R1 and R2 of Section 5. Notable contributions include methodological advancements such as the semantic injection of Mel-VAE++, a speech length predictor, and architectural ablations involving long-skip connections, offering actionable guidance for optimizing the promising modeling paradigm.

---

> > ### Author Response · Authors · 2024-11-19
> > **Authors' Response to Reviewer jyAn (2/2)**
> >
> > `[Q1] Request for statistical significance in result comparisons`
> >
> > > 1. Can you provide confidence intervals or p-values for the comparisons in your result tables? Without these, it is impossible to tell if the result is significantly better than state-of-the-art baselines.
> > >
> >
> > Thanks for your valuable point. We would like to note that our evaluation results are in line with the conventions of VALL-E and CLaM-TTS. While SPEAR-TTS included confidence intervals for CER, most previous studies, including all baselines in our work, do not report confidence intervals or p-values for objective metrics. For example, NaturalSpeech 2 and 3 do not include confidence intervals for any of their reported metrics.
> >
> > In response to your question, however, we calculated confidence intervals for Tables 1 and 2, with the updated results shown below. The small variances observed suggest that our results are generally reliable. We hope this addresses your concerns.
> >
> > Table 1 with CI
> > | Model         | WER ↓          | CER ↓          | SIM-o ↑         | SIM-r ↑         |
> > |---------------|----------------|----------------|-----------------|-----------------|
> > | DiTTo-en-S    | 2.01 ± 0.076   | 0.60 ± 0.020   | 0.4544 ± 0.0037 | 0.4935 ± 0.0037 |
> > | DiTTo-en-B    | 1.87 ± 0.073   | 0.52 ± 0.019   | 0.5535 ± 0.0035 | 0.5855 ± 0.0035 |
> > | DiTTo-en-L    | 1.85 ± 0.073   | 0.50 ± 0.018   | 0.5596 ± 0.0036 | 0.5913 ± 0.0035 |
> > | DiTTo-en-XL   | 1.78 ± 0.071   | 0.48 ± 0.018   | 0.5773 ± 0.0036 | 0.6075 ± 0.0035 |
> > | DiTTo-en-XL†  | 1.80 ± 0.072   | 0.48 ± 0.018   | 0.6051 ± 0.0035 | 0.6283 ± 0.0034 |
> >
> > Table 2 with CI
> > | Model         | WER ↓          | CER ↓          | SIM-o ↑         | SIM-r ↑         |
> > |---------------|----------------|----------------|-----------------|-----------------|
> > | DiTTo-en-S    | 3.07 ± 0.093   | 1.08 ± 0.027   | 0.4984 ± 0.0032 | 0.5373 ± 0.0031 |
> > | DiTTo-en-B    | 2.74 ± 0.088   | 0.98 ± 0.025   | 0.5977 ± 0.0030 | 0.6281 ± 0.0029 |
> > | DiTTo-en-L    | 2.69 ± 0.088   | 0.91 ± 0.025   | 0.6050 ± 0.0029 | 0.6355 ± 0.0028 |
> > | DiTTo-en-XL   | 2.56 ± 0.085   | 0.89 ± 0.024   | 0.6270 ± 0.0029 | 0.6554 ± 0.0028 |
> > | DiTTo-en-XL†  | 2.64 ± 0.086   | 0.94 ± 0.025   | 0.6538 ± 0.0028 | 0.6752 ± 0.0027 |
> >
> > `[Q2] Clarification on CMOS evaluation criteria for naturalness and intelligibility`
> >
> > > 2. In Appendix A3, the instruction for comparative mean opinion score (CMOS) shows that you only ask participants to compare the quality of the synthesized audio and reference, but in the main text you mention that DiTTo-en significantly outperforms the baseline models in naturalness, quality, and intelligibility (as assessed by CMOS). This raises questions about how the evaluations of naturalness and intelligibility for the synthesized audio were conducted in the experiments.
> > >
> >
> > We appreciate your thoughtful feedback. We assessed intelligibility primarily through WER and CER, as shown in Table 1, and further verified it using CMOS alongside naturalness. As detailed in the task instructions for CMOS in Figure 4 of our paper ("This assessment takes into account factors such as naturalness, clarity, and comprehensibility of the synthesized speech"), participants were instructed to evaluate audio quality while considering naturalness, clarity, and comprehensibility, with the latter two encompassing intelligibility.

---

> > > ### Author Response · Authors · 2024-11-24
> > > **Dear Reviewer jyAn**
> > >
> > > We sincerely thank you once again for your insightful feedback. Through our responses, we have aimed to address your concerns, and we would be deeply grateful if you could kindly review them. Please let us know if there are any additional points we can clarify to assist you in evaluating our paper.

---

> ### Author Response · Authors · 2024-11-30
> **Dear Reviewer jyAn**
>
> We appreciate the valuable feedback you have provided, which has helped us refine our work. With the discussion period now extended, we hope this provides an opportunity to understand whether our previous responses have adequately addressed the concerns you raised and to engage with any remaining questions or feedback. We deeply value your insights and are happy to provide further clarification or additional details to assist in your evaluation. Thank you for your engagement.

---

### Official Review · Reviewer_7SkZ · 2024-11-08

**Soundness:** 4
**Presentation:** 3
**Contribution:** 3
**Rating:** 8
**Confidence:** 5

**Summary:**

This work proposes DiTTo-TTS, a DiT-based architecture for zero-shot text-to-speech (TTS) synthesis applications. The proposed method does not rely on explicit alignments or phoneme units, thereby simplifying the TTS pipeline. For the neural codec, the paper proposes fine-tuning MelVAE to better align with pre-trained text encoders. This work includes exhaustive ablations on data and model scaling, architectural choices such as skip connections, choice of text encoders, and variable vs. fixed-length modeling to understand better the factors contributing to DiTTo-TTS's results. The authors also introduce a novel text and speech-token conditional speech length predictor based on auto-regressive training for variable-length modeling. Experiments on multilingual and English setups provide a detailed understanding of each component and demonstrate that DiTTo-TTS achieves SOTA results for zero-shot TTS without relying on domain-specific factors.

**Strengths:**

- The paper is clear and generally well-written. Detailed ablations highlight the importance of each factor in DiTTo-TTS.
- The contributions related to MelVAE fine-tuning and the auto-regressive speech length predictor are novel. While DiT has been explored previously in audio, the detailed ablations and contributions, such as long-skip connections, provide valuable insights for the field.
- The paper performs an exhaustive comparison against a number of baselines to demonstrate performance differences and establish SoTA results.

**Weaknesses:**

- No major weaknesses, there are some minor details which could be added. See comments section.

**Questions:**

- Table 7:
  - Are the DiTTo-Encodec and DiTTo-DAC models also fine-tuned with an LM objective? If not, please include DiTTo-MELVAE for comparison.
  - Please report sim-o when comparing different neural codecs.
- In general, reporting sim-o for other ablations would be beneficial for the community to facilitate comparisons with any results in the paper.
- Page 7, line 518: "We cannot measure the total inference time due to the lack of a speech length predictor for their embeddings, but even with ground truth lengths, their 7-8 times longer latents lead to significantly slower generation."
  - What is the feature rate for the MelVAE model? Encodec operates at 75Hz, so does this mean that MelVAE operates at approximately 10Hz? The original MelVAE in CLAM-TTS mentions a coderate of 100Hz. It would be helpful to include this information in the paper to make it more self-contained.
- How does speech editing, particularly for the infilling task, work? Do you only provide the prefix as a prompt to determine the duration of the edited segment? Specifically, could you describe the process for editing:

  **<prefix> <old text> <postfix> to <prefix> <new text> <postfix>**?
  - My guess is that this might require an alignment model to identify the start and end timings for <old text>, followed by predicting the duration of <prefix> <new text> conditioned on prefix speech tokens.
  - What happens if <prefix> is null? How does the model handle durations in this case? I assume with an alignment tool one could use any partial text and corresponding tokens for prompting.
- Ablation suggestion: Could you explore the impact of the number of training steps on performance?
- Minor:
  - Typo on page 1: "honemes".

---

> ### Author Response · Authors · 2024-11-19
> **Authors' Response to Reviewer 7SkZ (1/2)**
>
> We do appreciate the reviewer's positive feedback and constructive suggestions to further improve our paper. We hope our response thoroughly addresses your questions.
>
> Before we begin, we would like to clarify that the speech length predictor is trained with causal masking but generates the total length in a single forward pass during inference, making it non-autoregressive. Causal masking during training ensures that the predictor calculates the total length based solely on the given speech prompt, as described in the Speech Length Predictor paragraph of Section 3.2. Furthermore, the predictor is designed to simultaneously learn the remaining lengths for all positions within an audio sample, eliminating constraints on prompt length and enhancing training efficiency. This clarification has been added to the Inference paragraph in Appendix A.7 of the revised manuscript.
>
> `[Q1,Q2] Table 7: fine-tuning details & adding SIM-o results`
>
> > - Table 7:
> >     - Are the DiTTo-Encodec and DiTTo-DAC models also fine-tuned with an LM objective? If not, please include DiTTo-MELVAE for comparison.
> >     - Please report sim-o when comparing different neural codecs.
> > - In general, reporting sim-o for other ablations would be beneficial for the community to facilitate comparisons with any results in the paper.
>
> We thank the reviewer for insightful comments and valuable suggestions. We would like to clarify that both DiTTo-*encodec* and DiTTo-*dac* models In Table 7 are not fine-tuned with the LM objective. Also, DiTTo-*mls* (DiTTo-MELVAE) model is not fine-tuned with the LM objective as well.
>
> Initially, we observed that SIM-o and SIM-r followed similar trends in Tables 1 and 2, which led us to use SIM-r as a representative measure in subsequent ablation studies. As per your great suggestion, we have included SIM-o results in Table 7 of the revised manuscript as shown below (only codec-related results are included here).
>
> | Model            | WER ↓    | SIM-o ↑ | SIM-r ↑  |
> |------------------|----------|---------|----------|
> | DiTTo-*mls*        | 2.93     | 0.5467  | 0.5877   |
> | DiTTo-*encodec*    | 4.19     | 0.5105  | 0.5460   |
> | DiTTo-*dac-24k*    | 7.21     | 0.5478  | 0.5545   |
> | DiTTo-*dac-44k*    | 14.58    | 0.5391  | 0.5597   |
>
> DiTTo-*dac-24k* shows slightly higher SIM-o compared to DiTTo-*mls*. However, our preliminary experiments revealed a trade-off between WER and SIM, as shown in Table 22 of the revised manuscript. Specifically, reducing the scale factor from 0.3 to 0.2 resulted in a WER improvement from 7.21 to 4.70 (though still suboptimal), while SIM-o decreased from 0.5478 to 0.5302. We also include the discussion on this in the Appendix A.19 of the revised manuscript.

---

> > ### Author Response · Authors · 2024-11-19
> > **Authors' Response to Reviewer 7SkZ (2/2)**
> >
> > `[Q3] Clarification on Mel-VAE feature rate`
> >
> > > - Page 7, line 518: "We cannot measure the total inference time due to the lack of a speech length predictor for their embeddings, but even with ground truth lengths, their 7-8 times longer latents lead to significantly slower generation."
> > >     - What is the feature rate for the MelVAE model? Encodec operates at 75Hz, so does this mean that MelVAE operates at approximately 10Hz? The original MelVAE in CLAM-TTS mentions a coderate of 100Hz. It would be helpful to include this information in the paper to make it more self-contained.
> >
> > We appreciate your detailed feedback. In Section 7 of CLaM-TTS, it is stated that “Our approach enjoys a 10Hz codeword rate for efficient modeling.” Following your suggestion, we have added the following clarification to the section introducing Mel-VAE in Section 3.3 of the revised manuscript: "..., resulting in a 10.76 Hz code with high audio quality." This detail is also included in the Neural Audio Codec paragraph of Appendix A.1. Thank you for helping improve the clarity of our paper.
> >
> > `[Q4] Clarification on process and challenges in speech editing`
> >
> > > - How does speech editing, particularly for the infilling task, work? Do you only provide the prefix as a prompt to determine the duration of the edited segment? Specifically, could you describe the process for editing:
> > >
> > >     **<prefix> <old text> <postfix> to <prefix> <new text> <postfix>**?
> > >
> > >     - My guess is that this might require an alignment model to identify the start and end timings for <old text>, followed by predicting the duration of <prefix> <new text> conditioned on prefix speech tokens.
> > >     - What happens if <prefix> is null? How does the model handle durations in this case? I assume with an alignment tool one could use any partial text and corresponding tokens for prompting.
> >
> > Thanks for your question that gives us the opportunity to clarify our approach. In the speech editing demo, we need to know the start and end times of the segment to be edited within the target audio. We did not use the Speech Length Predictor for this task. We replace only the portion to be edited with noise, and the rest of the process remains identical to the inference procedure. The unedited segments serve as speech prompts for the edited portion, and the same text is conditioned in the same manner.
> >
> > Your suggestion to automate the selection of the segment to edit is magnificent, and we appreciate it. We have experimented with using Whisper to automatically detect the end time of the word preceding the target word and the start time of the word following it, which proved to be quite useful. This automation enhances the editing process, and we thank you for bringing this idea to our attention.
> >
> > `[Q5] Suggestion: Ablation on impact of training steps on performance`
> >
> > > - Ablation suggestion: Could you explore the impact of the number of training steps on performance?
> >
> > As per your great suggestion, we conducted evaluations to examine the impact of training steps on performance. Specifically, we assessed the *cross-sentence* task performance of the DiTTo-*mls* model at 50K, 100K, 150K, and 200K training steps to understand training dynamics. The results are summarized in the table below. Our findings indicate that the DiTTo-*mls* model reaches reasonable WER and SIM performance as early as 100K steps. We have added the suggested ablation to Appendix A.18 (IMPACT OF TRAINING STEPS ON PERFORMANCE) in the revised manuscript. We will also add the results for the DiTTo-en-XL model soon. Please note that SIM-o is reported here solely to show its correlation with SIM-r.
> >
> > | Training Steps   | WER ↓ | SIM-o ↑ | SIM-r ↑ |
> > |---------|-------|---------|---------|
> > | *50K*   | 7.27  | 0.4697 | 0.5225 |
> > | *100K*  | 3.51  | 0.5182 | 0.5630 |
> > | *150K*  | 3.17  | 0.5342 | 0.5770 |
> > | *200K* (from Table 4 of our paper)  | 2.93  | 0.5467 | 0.5877 |
> >
> > We sincerely thank you once again for your valuable suggestion and greatly appreciate your contribution to enhancing the quality of our paper.
> >
> > `[Q6] Typo`
> >
> > > - Minor:
> > >     - Typo on page 1: "honemes".
> >
> > Thank you for spotting the typo. We have corrected it to "phonemes."

---

> > > ### Comment · Reviewer_7SkZ · 2024-11-19
> > > **Thanks for explanations.**
> > >
> > > I am satisfied with the clarifications and will maintain my scores. I do have one further comment.
> > > > In the speech editing demo, we need to know the start and end times of the segment to be edited within the target audio. We did not use the Speech Length Predictor for this task.  We replace only the portion to be edited with noise, and the rest of the process remains identical to the inference procedure.
> > >
> > > To me this would seem like a limitation as replacing an old text with a new one when there lengths significantly differ can introduce unnaturalness. Like mentioned above, using an ASR model for alignment should be able to fix this pretty easily.

---

> > > > ### Author Response · Authors · 2024-11-19
> > > > **Re: Thanks for explanations.**
> > > >
> > > > We are pleased to hear that our clarifications have satisfied the reviewer. We agree with your suggestion, and a straightforward solution would be to manually adjust the length of the newly generated speech segments in proportion to the length of the new text. We sincerely thank the reviewer once again for your efforts in evaluating our work and helping us improve the paper.

---

### Official Review · Reviewer_sHU2 · 2024-11-08

**Soundness:** 3
**Presentation:** 3
**Contribution:** 2
**Rating:** 6
**Confidence:** 4

**Summary:**

This paper introduces a latent diffusion model based zero-shot TTS as DiTTo-TTS.  The author have done rigorous investigations to improve LDM based TTS by replacing U-Net with DiT, remove dependency of domain-specific factors like phoneme and duration with several experiments on the model size and datasets. The experiment results shows the good performance of naturalness (CMOS), intelligibility (CER/WER) and similarity (SMOS and SIM).

The major achievements prove that the LDM TTS model could work as end-to-end in a joint modeling way for the text and speech representation, and it is scalable to different model size and dataset.

**Strengths:**

This paper clearly describes the work for an end-to-end fully diffusion model for zero-shot TTS. The strengths lie on the following aspects
1) Rich experiments have done to investigate the different model size of DiTTo model (small, base, large and Xlarge), data scalability; comparison between U-Net and DiT module, and comparison between speech length model with variable and fixed and etc.
2) A lot of state-of-art TTS system comparison is provided. Although not all the system have compared all the metrics but several metrics are showed for readers to compare and get a relative complete picture on the metrics of different popular system include AR model like VALL-E, CLaM-TTS, NAR model like YourTTS, and SimpleTTS,  etc.
3) Ablation study and demo page quit clear.

**Weaknesses:**

The major weakness of this paper is the novelty. The most contribution as the author also mentioned is removing the domain-specific factors like phoneme and duration, and just giving a total duration estimation is good enough for the task which is even better. However, this is not new, and similar work has been done at SeedTTS https://arxiv.org/pdf/2406.02430#page=6&zoom=100,144,548.

SeedTTS-DiT has no-dependence on the phoneme duration by just giving the total duration for the diffusion model. It already shows that this fully diffusion based speech generation model could achieve superior performance. This integration is just a quite natural way regarding the highly flexibility of noise iteration with DiT structure for a diffusion or flowmatching model. I would believe many folks/peers have applied it in speech generation before or after Seed-TTS-DiT and it indeed works as expected. According to these information, I would give the question on the novelty of this paper.

**Questions:**

1. Regarding successfully applying joint latent diffusion model for TTS,  the comparison between the autoregressive (AR) model like a language speech model (CLaM in this paper) and diffusion model would be attracting as they are the current two major methodologies as diffusion vs. AR model.  According to the result, the diffusion model archives better similarity and quality from subjective metrics, and also better CER. However, as the total training data is not large enough (82K), would it be enough to be convincing that diffusion model could achieve superior result rather than AR model while language model may work better on larger dataset? It would be a fantastic topic to discuss.
2. The framework design a latent representation for diffusion model to generate, then use Mel-VAE to decode latent into mel-spectrogram, and get the synthesized speech via BigVGAN. This is a little confusing why not just model mel-spectrogram for diffusion model?
3. The Table 3 shows the subjective result with SMOS and CMOS. How many speakers and how many test utterances in this test set? It would be related to the diversity of the model  capability.

---

> ### Author Response · Authors · 2024-11-19
> **Authors' Response to Reviewer sHU2 (1/2)**
>
> We sincerely appreciate your constructive feedback and thoughtful suggestions, which have been invaluable in improving the manuscript. Below, we address your comments point by point.
>
> `[W1] Regarding contributions and the novelty of the work`
>
> > The major weakness of this paper is the novelty. The most contribution as the author also mentioned is removing the domain-specific factors like phoneme and duration, and just giving a total duration estimation is good enough for the task which is even better. However, this is not new, and similar work has been done at SeedTTS https://arxiv.org/pdf/2406.02430#page=6&zoom=100,144,548.
> >
> >
> > SeedTTS-DiT has no-dependence on the phoneme duration by just giving the total duration for the diffusion model. It already shows that this fully diffusion based speech generation model could achieve superior performance. This integration is just a quite natural way regarding the highly flexibility of noise iteration with DiT structure for a diffusion or flowmatching model. I would believe many folks/peers have applied it in speech generation before or after Seed-TTS-DiT and it indeed works as expected. According to these information, I would give the question on the novelty of this paper.
> >
>
> We thank the reviewer for thoughtful feedback regarding the novelty of our work. We would like to respectfully argue that the removal of domain-specific factors such as phonemes and duration is not positioned as our main contribution. As noted in the paper, we acknowledge prior pioneering works (Simple-TTS and E3 TTS) that have explored these ideas and highlight their limitations to contextualize our study.
>
> We also acknowledge the concurrent work Seed-TTS, which demonstrates impressive performance with a simple modeling approach. However, Seed-TTS does not provide a comprehensive comparison with diverse baselines (**RQ1**) and the critical factors that enable LDMs to succeed without domain-specific factors (**RQ2**), which are the central focus of our study as discussed in Section 5:
>
> - **RQ1:** Can LDM achieve state-of-the-art performance in text-to-speech tasks at scale without relying on domain-specific factors, similar to successes in other domains?
> - **RQ2:** If LDM can achieve this, what are the primary aspects that enable its success?
>
> Our primary contribution lies in systematically addressing these questions through extensive empirical investigations. We demonstrate that LDM-based TTS, even without domain-specific factors, can achieve performance that is superior or comparable to various state-of-the-art autoregressive (AR) and non-autoregressive (Non-AR) TTS baselines, enabling a fair evaluation of its performance within the broader research landscape. Additionally, we identify the key factors driving this success and provide practical insights into a new modeling paradigm for TTS systems. Notable contributions include methodological advancements such as the semantic injection of Mel-VAE++, a speech length predictor, and architectural ablations involving long-skip connections, offering actionable guidance for optimizing LDM-based TTS systems. While the technical novelty in model architecture design may appear incremental, we believe our work makes a significant contribution to advancing the understanding of this emerging paradigm and shaping the future direction of TTS modeling.
>
> We have incorporated this discussion into Appendix A.1 under Related Work in the revised manuscript and sincerely thank you for your valuable contribution to enhancing the clarity and quality of our paper.

---

> > ### Author Response · Authors · 2024-11-19
> > **Authors' Response to Reviewer sHU2 (2/2)**
> >
> > `[Q1] AR vs. Diffusion model on larger dataset`
> >
> > > 1. Regarding successfully applying joint latent diffusion model for TTS, the comparison between the autoregressive (AR) model like a language speech model (CLaM in this paper) and diffusion model would be attracting as they are the current two major methodologies as diffusion vs. AR model. According to the result, the diffusion model archives better similarity and quality from subjective metrics, and also better CER. However, as the total training data is not large enough (82K), would it be enough to be convincing that diffusion model could achieve superior result rather than AR model while language model may work better on larger dataset? It would be a fantastic topic to discuss.
> > >
> >
> > Thanks for your insightful comments and for highlighting this interesting topic. We agree that exploring performance comparison of two models on larger scale dataset would be a fascinating topic to discussion and the direction for future research. Larger-scale diffusion-based models like Seed-TTS—which leverage datasets orders of magnitude larger than previous TTS systems, such as the 200K hours used in NaturalSpeech 3—consistently demonstrate superior or comparable performance to next-token prediction models in terms of speaker similarity and WER. This indicates that diffusion models may maintain their advantages even with larger datasets. Nonetheless, we acknowledge that exploring the performance of these models on much larger datasets, potentially approaching the scale of models like GPT—though building such a large dataset in the speech domain presents its own challenges—would be a fascinating direction for future research.
> >
> > `[Q2] Rationale for using latent representation instead of mel-spectrogram in diffusion modeling`
> >
> > > 2. The framework design a latent representation for diffusion model to generate, then use Mel-VAE to decode latent into mel-spectrogram, and get the synthesized speech via BigVGAN. This is a little confusing why not just model mel-spectrogram for diffusion model?
> > >
> >
> > Thanks for your question that gives us the opportunity to clarify our framework. We chose to use a latent representation from Mel-VAE instead of directly modeling mel-spectrograms because compact representations in terms of time-domain improve diffusion model performance, particularly in intelligibility, as demonstrated by the WER results in Table 23 of the revised manuscript. Compressing mel-spectrograms into a latent space enables more efficient modeling of speech features.
> >
> > To support this, we conducted additional experiments comparing DiTTo-*mel* (trained on mel-spectrograms) with DiTTo-*encodec* and DiTTo-*mls* (trained on latent representations). In the table below, the results indicate that greater compression (DiTTo-*mls* > DiTTo-*encodec* > DiTTo-*mel*) improves WER performance by shortening sequence length in the time domain, consistent with our hypothesis in Appendix A.20.
> >
> > | Model                      | WER ↓ | SIM-o ↑ | SIM-r ↑ |
> > |----------------------------|-------|---------|---------|
> > | DiTTo-*mls* (from Table 7)    | 2.93 | 0.5467 | 0.5877 |
> > | DiTTo-*encodec* (from Table 7)| 4.19 | 0.5105 | 0.5460 |
> > | DiTTo-*mel*                   | 4.65 | 0.5637 | 0.5792 |
> >
> > DiTTo-*mel* achieves the highest SIM-o score, as directly predicting ground truth mel-spectrograms and avoiding the information loss typically introduced by autoencoder latent decoding. However, the overall performance of DiTTo-*mls* is comparable when considering both SIM-o and SIM-r together, while showing noticeably better intelligibility than DiTTo-*mel*. As shown in Table 1 and Section 5.1, DiTTo-*mls* also offers fast inference speeds and remains faster than Voicebox that directly models mel-spectrograms, with the same number of diffusion steps. This highlights the effectiveness of Mel-VAE as a target latent representation, balancing intelligibility, speech similarity, and efficiency.
> >
> > `[Q3] Clarification on test set details for SMOS and CMOS evaluation`
> >
> > > 3. The Table 3 shows the subjective result with SMOS and CMOS. How many speakers and how many test utterances in this test set? It would be related to the diversity of the model capability.
> > >
> >
> > We used 40 test utterances from the LibriSpeech test-clean set, specifically sampling one utterance from each of the 40 speakers. We have added this information to the caption of Table 3 of the revised manuscript. Thank you for helping us improve the clarity of our work.

---

> > ### Comment · Reviewer_sHU2 · 2024-11-22
> >
> > Thanks for the detail explanation and clarification which help me understand more about the paper's contribution and primary goal.   Regarding the RQ1 and RQ2, the paper has demonstrated good perspective and comprehensive experiment for LDM-based TTS, which is quite meaningful to both research and industry community.

---

> ### Comment · Reviewer_sHU2 · 2024-11-22
>
> Thanks the authors' fruitful discussion and additional experiment for the clarify.
>
> For the Q1, it is quite encouraging to extend more experiments to explore the capability of diffusion model for speech generation. Is fully diffusion model for speech generation better or comparable as LLM? and each model's advantages and limitations for now or then.
>
> For the Q2, I think this is very interesting result. As the direct thought would be modeling mel, however, the result shows even with mel as intermediate feature for vocoder, it is still useful to adopt latent representation for LDM model and decoder as mel then. Have you conducted any subjective test on the quality and similarity to valid the table objective result? It would be quite helpful if these result include in the paper as well.
>
> At last, I would change the total rating score from 5 -> 6 according to authors detail reply especially for the contribution and primary goal of the paper, but still keep the each item score especially for the novelty.

---

> > ### Author Response · Authors · 2024-11-24
> > **Re: Official Comment by Reviewer sHU2**
> >
> > The authors sincerely thank the reviewer for their insightful feedback and positive evaluation.
> >
> > Regarding **Q1**, we appreciate your encouraging comments on exploring the capabilities of fully diffusion-based models compared to LLMs. We plan to investigate this further in our future work.
> >
> > For **Q2**, based on your recommendation, we conducted subjective evaluations using the same methodology as described in Section 5.1. The results are shown in the table below.
> >
> > Both SMOS and CMOS subjective evaluations confirm that DiTTo-*mls* outperforms the other models, validating the objective results. The notably poor performance of DiTTo-*encodec* appears to stem from the relatively lower quality of the EnCodec. As suggested, we have included this analysis and the results in Appendix A.21 of the revised manuscript.
> >
> > | Model                      | WER ↓ | SIM-o ↑ | SIM-r ↑ | SMOS | CMOS |
> > |----------------------------|-------|---------|---------|---------|---------|
> > | DiTTo-*mls* (from Table 7)    | 2.93 | 0.5467 | 0.5877 | 3.67±0.14 | 0.00 |
> > | DiTTo-*encodec* (from Table 7)| 4.19 | 0.5105 | 0.5460 | 2.86±0.15 | -1.43 |
> > | DiTTo-*mel*                   | 4.65 | 0.5637 | 0.5792 | 3.58±0.14 | -0.44 |
> >
> > Once again, we sincerely thank the reviewer for their detailed feedback and recognition of the contributions and primary goals of our paper. We are happy to address any further questions or suggestions if you have.

---

> > > ### Comment · Reviewer_sHU2 · 2024-12-02
> > >
> > > Thank you for your detail response and experiment. I'm glad to see the results and changes, which clear my doubts and make the paper even more insightful. I have no other questions, and thanks the authors.

---

### Author Response · Authors · 2024-11-19
**General Response**

(R1 = R-sHU2, R2 = R-7SkZ, R3 = R-jyAn, R4 = R-Tbzi)

We sincerely thank the reviewers for their constructive feedback and the time and effort they dedicated to evaluating our work. We appreciate the reviewers that acknowledge the novelty of our contributions (R2), the strong performance of DiTTo-TTS (R2, R3), our exhaustive comparisons, extensive ablations, and the clarity of our paper (R1, R2, R4). We provide detailed responses to each point in our individual replies and believe we address all concerns and questions raised by the reviewers. Based on this feedback, we revise the manuscript, with modifications highlighted in blue.

`Addressing Novelty Concerns`

We would like to revisit and address the most common concern raised by the reviewers regarding the novelty of our work as part of our general response.

- Our primary contribution lies not in introducing new techniques but in conducting an experimental analysis of the key aspects that enable LDM-based TTS without domain-specific factors, such as phonemes and phoneme-level durations, to achieve state-of-the-art performance, as validated by our final model's strong results.
- Specifically, we provide a comprehensive comparison with diverse baselines (**RQ1**) and explore the critical factors that enable LDM-based TTS to succeed without domain-specific factors (**RQ2**):
    - **RQ1:** Can LDM achieve state-of-the-art performance in text-to-speech tasks at scale without relying on domain-specific factors, similar to successes in other domains?
    - **RQ2:** If LDM can achieve this, what are the primary aspects that enable its success?
- In exploring these questions, we introduce methodological advancements such as the semantic injection of Mel-VAE++, a speech length predictor, and architectural ablations involving long-skip connections, providing unique insights and actionable guidances for optimizing the promising modeling paradigm.

We hope this clarifies the significance of our contributions and addresses the reviewers' concerns. We believe our work complements existing research, enhancing both the performance and understanding of LDM-based TTS without domain-specific factors.

---

### Meta-Review · Area_Chair_2mMY · 2024-12-07

**Metareview:**

The paper includes extensive experiments to explore different model sizes (small, base, large, Xlarge), data scalability, and comparisons between U-Net and DiT modules, as well as variable and fixed speech length models.  It provides comparisons with various state-of-the-art TTS systems, including AR models like VALL-E and CLaM-TTS, and NAR models like YourTTS and Simple-TTS, across several metrics. The ablation studies and demo page are well-presented, highlighting the importance of each factor in DiTTo-TTS. The paper is clear and generally well-written, making it easy to understand the detailed experiments and findings.

**Additional Comments On Reviewer Discussion:**

The paper's novelty is limited as similar work has been done in SeedTTS, and it relies on pre-trained components, raising questions about its independence from domain-specific factors. Additionally, the approach of using pre-trained text encoders and neural audio codecs has been seen in previous works like NaturalSpeech 2. Despite these issues, the systematic analysis and guidance on design choices are valuable contributions. These issues have been addressed during the author-reviewer discussion.

---

### Decision · Program_Chairs · 2025-01-22

Accept (Poster)